# A Sustainable Port-Hinterland Container Transport System: The Simulation-Based Scenarios for CO$_2$ Emission Reduction

Khandaker Rasel Hasan ⓘ, Wei Zhang ⓘ and Wenming Shi *ⓘ

Centre for Maritime and Logistics Management, Australian Maritime College, University of Tasmania, 100 Maritime Way, Newnham, TAS 7248, Australia; khandaker.hasan@utas.edu.au (K.R.H.); vera.zhang@utas.edu.au (W.Z.)
* Correspondence: wenming.shi@utas.edu.au

**Abstract:** This paper calculates the CO$_2$ emissions for the port-hinterland container transport system and proposes possible emission reduction measures. This paper considers the Dhaka–Chittagong port-hinterland transport system in Bangladesh. The port-hinterland transport system represents 70% of the total international maritime containerised trade, including more than 2.0 million twenty-foot equivalent units (TEUs) per year. By implementing different scenarios using a simulation approach, this research suggests a substantial reduction in CO$_2$ emissions for the port-hinterland transport system. The scenarios include infrastructure development and performance and operational efficiency improvement in the port and modal shift for the hinterland. In formulating the scenarios, the current performance statistics of the port and its hinterland as well as the possibility of the implementation of these scenarios are carefully analysed. The findings depict that Bangladesh could significantly contribute to the reduction in port-hinterland CO$_2$ emissions by implementing the suggested scenarios.

**Keywords:** CO$_2$ emission reduction; port-hinterland container transport system; simulation-based scenario





## 1. Introduction

Sustainability comprises the environmental, social and economic dimensions of ports. The environmental perspective is associated with noise pollution, air quality, dredging operation and dredging disposal. The economic perspective is very much concerned with financial performance, operational performance and investment for further performance enhancement. The social perspective covers the direct and indirect contributions to employment, the port–city interaction and relationship, liveability at the port neighbourhoods and the contribution to education and knowledge development [1,2]. Among these various dimensions of sustainability, CO$_2$ emission reduction directly affects the air quality, the liveability of the area surrounding the port and the operational efficiency and indirectly affects the other dimensions [3]. For example, by creating more jobs in port-related industries and more operational activities, ports are enabling more travel for people, which is indirectly linked with the port operation and port-related CO$_2$ emissions. As a result, CO$_2$ emissions from shipping operations are receiving greater emphasis as their contribution to global carbon emissions is expected to rise in the coming years. Even though shipping activities account for 2.4% of global CO$_2$ emissions, it is believed that this will be tripled by 2050 [3]. Ports, as the central stakeholder of shipping activities, have a greater role to play from the positions of regulator and operator. The main sources of carbon emissions within the port include the vessels calling the port, cargo handling equipment and inland cargo movement using different modes. Ports can play a role in all these areas by implementing different regulatory, monetary and operational measures to reduce emissions.

There is evidence that not only operational costs but also CO$_2$ emissions are influenced by operational efficiency in ports [4]. However, a port performance evaluation considering

the sustainability aspects proposed by Castellano and Ferretti [5] found that ports with the largest performance improvement in Italy are also the highest GHG emitters and that, at the same time, these ports had the port authorities who were most committed to adopting sustainable practices. This explains the notion that operational performance improvement means more operational activities, leading towards more carbon emissions. However, this is from the perspective of total emissions, and the emissions per container or per tonnage could show a decline. Another important aspect is the environmental impact of hinterland transport, which is basically an extension of the port services in which the actual producers or consumers are originated or destined. Therefore, sustainable port-hinterland connectivity can directly promote ports to claim their sustainable profiles [6]. Geerlings [7] pointed out that sustainability in the transport sector can be achieved in four ways, comprising three short-term actions and one long-term action. The short-term actions include (i) a reduction in the impact of modes by technological means, (ii) shifting to less damaging modes of transport and (iii) a reduction in the total amount of transport undertaken. The long-term action is the improvement of spatial planning, which can reduce the distance between activities and lower the need for mobility. Similarly, these actions can also be applied to achieve port sustainability.

In this paper, $CO_2$ emissions in Chittagong port and its hinterland are calculated, and some scenarios are implemented via simulations that contribute to sustainability by reducing $CO_2$ emissions. Firstly, the approximate fuel consumption in various operational activities of Chittagong port, which are marine operations, terminal and yard operations, and gate operations, as well as in the hinterland transportation of the port are calculated. The fuel consumption data are then used for the calculation of total $CO_2$ emissions from the Chittagong port and its hinterland operation. Then, some scenarios are implemented via simulation to examine the impact on fuel consumption and $CO_2$ emission reduction. In the development of the scenarios, the aspects of infrastructure, operational efficiency and modal shifting are taken into consideration. The findings show that a substantial $CO_2$ emission reduction is possible in the case of Chittagong port and the Dhaka–Chittagong hinterland transport. As the 58th largest container port in the world in terms of throughput and the main gateway port in Bangladesh, Chittagong port handles around three million TEUs annually and more than 90% of Bangladesh's containerised international trade. Notably, the Dhaka–Chittagong hinterland transports around 70% of these containers. As a result, any $CO_2$ emission reduction in the case of the port-hinterland container transport system can significantly impact the local emission scenarios. On the other hand, the existing port-related literature has not paid sufficient attention to port hinterlands from an environmental sustainability perspective. Referring to a recent study on the competitiveness of this port, users suggest that the environmental practices have the least competitive aspect for port attractiveness [8]. Therefore, the calculation of $CO_2$ emissions and the proposed emission reduction solutions via the implementation of different scenarios can contribute to the existing port management literature from an environmental sustainability perspective. In addition, this research provides implications for examining port users' $CO_2$ emissions while conducting their business, thereby contributing to port service providers and policymakers when formulating sustainability policies for port-hinterland transportation.

The paper begins by briefly discussing the literature concerning port-hinterland $CO_2$ emissions, the Chittagong port and the Dhaka–Chittagong corridor and their performance data in Section 2. The performance data are the main input data for calculating $CO_2$ emissions and designing and implementing various scenarios via simulation analysis. Then, the $CO_2$ emission calculation methods and materials are briefly presented in Section 3. In Section 4, the total $CO_2$ emissions in the Chittagong port and the Dhaka–Chittagong hinterland and the impact of the implementation of different scenarios and simulations on port emissions and modal shift on hinterland emissions are observed and discussed in detail. It is worth mentioning that Sections 2–4 include a respective data description and discussion with a consideration of data that is relevant to each section. Section 5 summarises the findings and the contributions of the paper, and Section 6 concludes the paper.

## 2. Literature Review

### 2.1. $CO_2$ Emissions in Port-Hinterland Transport System

In realising the current and future contributions of the maritime sector to global $CO_2$ emissions, much emphasis has been placed on the $CO_2$ emissions from shipping and port activities [3,9]. According to Wang and Peng [9], in the case of ports' $CO_2$ emissions, the focus was primarily on shipping activities and handling activities, while an increasing recent focus on the inland distribution system can also be noted in other studies [10]. For example, Chang and Song [11] estimated the fuel consumption of the container vessels in the case of Incheon port, and this estimation considered various sizes of vessels including the voice from the anchorage to the berth. This research only considers marine operations, focusing the location of the highest possible emissions and the types of the highest emitting vessels. On the other hand, Huzaifi and Budiyanto [12] calculated the $CO_2$ emitted by the Jakarta International Container Terminal, considering the services from berthing to removing containers from the port and mainly examining berthing and terminal operations but not the waiting time. This research focuses on the importance of terminal layout on $CO_2$ emissions. Another study by Geerlings and van Duin [13] concluded that terminal layout is one of the most effective methods of $CO_2$ emission reduction.

Emissions from the terminal equipment and the measures for reduction in the port of Valencia were analysed by Martínez-Moya and Vazquez-Paja [14]. They categorised the highest emitting equipment and suggested retrofitting the RTGs and introducing LNG powered tractors. Alternative energy and fuels were identified as important influential factors for decreasing energy consumption and carbon emissions by the IMO and other reviews [15]. Utilizing renewable energy sources for port equipment operations is an increasing area of interest in current academic research [16]. On the other hand, the scheduling problems of port operating equipment have been extensively observed in the existing literature. For example, the carbon efficient scheduling of rubber-tyred gantry cranes (RTGs) and electric rubber-tyred gantry cranes (ERTGs) was analysed by Chen and Zeng [17]. A specific focus on emission reduction by optimizing truck operations in the port areas was also observed from Chen and Govindan [18]. Their findings indicated that the reduction in truck waiting time and shifting in truck arrival time can significantly reduce the emissions.

Liao and Tseng [19] calculated the $CO_2$ emissions for Taiwanese ports and their inland distribution system and came up with the solution that changing a transhipment route for the inland distribution of the containers can reduce the $CO_2$ emissions. Shin and Cheong [20] calculated the emissions in Busan port for all the port-related equipment, including marine vessels, cargo handling equipment, heavy duty trucks and rail locomotives. They mainly located the highest emitting areas and the relation between emissions and the amount of cargo. The case of Qingdao port in China was analysed by Mamatok and Huang [21] via the implementation of scenario-based simulations. They argued that scenarios such as operating time optimization, spatial measures, equipment modernization and modal shift are important to $CO_2$ emission reduction. The port-hinterland container transport system including three ports was analysed by Li and Kuang [22] considering various scenarios on port selection based on new routes and multimodal transport as well as their impacts. In order to reduce carbon emissions in the port of Shenzhen, possible suggestions, such as improving loading and unloading efficiency, connecting to shore power and using low sulphur fuel oil. were proposed by Yang and Cai [23]. In parallel to some of the solutions for reducing $CO_2$ emissions from port-related activities that were addressed in the above discussion, Yun and Xiangda [24] suggested that the most studied measures used to reduce $CO_2$ emissions in port-related activities are onshore power supplies, reduced speed in water channels and reduced turnaround time. It is observed that all the measures implemented to reduce $CO_2$ emissions highlighted above broadly cover the aspects of operational efficiency and infrastructural development. Therefore, in the case of formulating the scenarios for Chittagong port, the aspects of operational efficiency and infrastructure development are taken into consideration.

### 2.2. $CO_2$ Emissions in Different Inland Transport Systems

The four methods for achieving sustainability in transport that were suggested by Geerlings [7] were further explained and elaborated by Hou and Geerlings [6] and the IPCC [25] from the perspectives of actions to be undertaken. Several studies considered these methods to attempt to significantly reduce carbon emissions from road transportation [26,27] because roads are still the predominating contributor of carbon emissions in the transport sector. For example, McKinnon and Piecyk [28] offered recommendations for the emissions factor in chemical transport operations in Europe, for which they emphasised modal split, restructuring the supply chain, improving vehicle utilization and fuel efficiency. Wang and Peng [9] showed a comparative fuel consumption and emissions factor for three different modes, namely, road, rail and inland ships, in China for the period from 2006 to 2011. They suggested that phasing out obsolete and aging equipment, installing energy saving devices and new energy saving equipment, optimizing the container distribution network, implementing multimodal transport, improving the logistics information sharing and implementing a clean truck program can be promising solutions for $CO_2$ emission reduction. A comparatively generic $CO_2$ emission calculation was conducted by the IPCC [25]. The report included an extensive discussion of $CO_2$ emission reduction pathways in which vehicle technology and propulsion systems and the behavioural aspects of accepting their use in real life, modal shifts, infrastructure, new routes, the relocation of production and the reconfiguration of the global supply chain were covered.

The comparative picture of $CO_2$ emissions from the above studies is presented in Table 1 below. Even though there are dissimilarities among the quantity of emissions for different modes in the literature, it provides an understanding of the degree of emissions among the different modes.

**Table 1.** Comparison of $gCO_2$/ton-km emissions factors in different studies.

| Modes | Type of Vehicle | McKinnon and Piecyk [28] | IPCC [25] | Wang and Peng [9] [1] |
|---|---|---|---|---|
| Road | Heavy duty | 59–109; average: 62 | 100–190 | 188.50 |
| | Medium | | 240–370 | |
| Rail | Heavy | 7.3–23; average: 22 | 18–25 | 8.30 |
| | Light | | 26–33 | |
| IWT | | 31 | | 18.85 |
| Barge | | | 10–50 | |
| Coastal container ship | | 14 | | |
| Rail–road | | 24–30; average: 26 | | |
| IWT–road | | 32–37; average: 34 | | |
| Shortsea–road | | 16–23; average: 19 | | |

Source: Compiled by the author based on McKinnon and Piecyk [28], IPCC [25] and Wang and Peng [9] ([1] According to this research, in 2011, rail transport consumed 26.40 kg of diesel to transport 10,000 ton-km, whereas road transport used 6 L of diesel to transport 100 ton-km and inland shipping consumed 6 kg of diesel to transport 1000 ton-km. In order to convert this consumption to $gCO_2$/ton-km, for example, the 26.40 kg of diesel to transport 10,000 ton-km using rail is multiplied by 1000 to convert the consumption from kg to gram and then divided by 10,000 to obtain the amount gram (g) per ton-km and, finally, multiplied by the emissions conversion factor for diesel, which is 3.141, to obtain the final emissions for rail transport, which is 8.3 $gCO_2$/ton-km = $((26.40 \times 1000)/10{,}000) \times 3.141$. Although the figure for road is given in litres, to convert litres to grams, the process is the same: multiply by 1000. Converting these data to the $CO_2$ emissions factor shows that the respective emissions for road, rail and inland shipping is approximately 188.5 $gCO_2$/ton-km, 8.3 $gCO_2$/ton-km and 18.85 $gCO_2$/ton-km).

Although the fuel consumption is very much associated with the type of vehicle, its age, the type of fuel in use and the infrastructure of the country of operation, the emission range observed in Table 1 for IWT barges or inland or coastal shipping show some sort

of similarities. The $gCO_2$/ton-km emissions suggested by McKinnon and Piecyk [28] and Wang and Peng [9] lies within the range of inland shipping suggested by the IPCC [25]. It is important to highlight that the lower level of emissions mostly for road and rail represents the vehicles having dual fuel or technically sound combusting facilities. In the case of the emissions figure for rail, the calculations by McKinnon and Piecyk [28] and the IPCC [25] show relatively close figures, whereas the findings from Wang and Peng [9] show very low emissions from rail. On the other hand, the emissions figure for road transport shows some similarities between the IPCC [25] and Wang and Peng [9] but a significant difference with McKinnon and Piecyk [28]. It is further observed that, as the carrying load increases, the $gCO_2$/ton-km starts declining, and this could be the reason behind a notable difference among the emission figures from these studies. McKinnon and Piecyk [28] showed that a carrying load for road from 10 tons to 30 tons reduces the emissions from 80 $gCO_2$/ton-km to 50 $gCO_2$/ton-km. In their emissions calculations, they have considered a typical 40 ton capacity truck with a carrying load of 26 tons. Moreover, McKinnon and Piecyk [29] highlighted the role of interruption in vehicle movement as an important factor for road transport fuel consumption. The study pointed out that a truck carrying a 40-ton load and achieving an average speed of 50 km/h would consume 28, 52 and 84 L of fuel per 100 km if it stopped zero, one and two times per km. Therefore, variability seems quite logical in the case of the emissions factor for different places of operation, vehicles and road conditions. The IPCC [25] study showed that if the load factor increases the per ton-km emissions reduces for rail too. As shown in Table 1, the rail emissions reduce from 26–33 $gCO_2$/ton-km to 18–25 $gCO_2$/ton-km if the load is heavy. Moreover, the emission calculation by IPCC (2014) [25] emphasised that the measure is very much indicative here as there are not many comprehensive studies covering the full range of vehicles and technologies that are available. Therefore, to sum up the above discussion, it is observed that the $CO_2$ emissions for different modes in different regions could show significant variations. Among others, the type of vehicle, its age, the type of fuel in use and the infrastructure of the country of operation are important factors to consider. In order to calculate the $CO_2$ emissions in the case of hinterland transport, therefore, the real data on these factors are very important. However, if not available, the overall ranges represented in the various studies can be used, at least, to have some indicative measurements of the emission differences among modes. In this study, the emissions factor for road transport is proposed as 180 $gCO_2$/ton-km, which is within and close to the upper limit for the heavy duty truck emissions calculated by the IPCC [25] and very close to the emissions factor calculated by Wang and Peng [9]. For rail, it is proposed as 26 $gCO_2$/ton-km, which is close to the upper limit of the range calculated by McKinnon and Piecyk [28] and almost same as the highest value and the lowest value of both the heavy- and light-duty rail emission ranges calculated by the IPCC [25]. On the other hand, in the case of IWT, the emissions factor is considered 34 $gCO_2$/ton-km, which is very similar to that of the IWT–road combination calculated by McKinnon and Piecyk [28] and within and close to the upper limit of the range calculated by the IPCC [25]. The proposed emissions factor for IWT is higher than the emissions factor calculated by Wang and Peng [9]. If we compare the proposed emissions factor with the different ranges identified from the literature and summarised in Table 1, it is observed that, for road, rail and IWT, they have close to the highest possible emissions considering their heavy-duty cargo, which can at least indicate the highest possible total emissions from the individual sectors.

### 2.3. Chittagong Port and Its Performance

Bangladesh is the 41st largest economy in the world and has shown an average GDP growth rate of around 6% in recent decades. Its share of international trade comprises 35% of Bangladesh's national economy. Being the second largest exporter of readymade garments in the world, the country is showing prospective growth in the garment-related sectors such as textile, knitwear and others. The growing import and export of garments is highly dependent on maritime transportation. The Chittagong port is the prime facilitator

of garment trade, handling more than 90% of the international maritime trade of the country. The port shows an average 9% growth in the case of total cargo in tons, 6% growth in the case of container throughput in TEUs and more than 7% growth in the case of the total number of vessel calls, as presented in Table 2.

**Table 2.** Throughput and Vessel Call.

| Year | Total MT | Growth | Total TEUs | Growth | Vessel Calls | Growth |
|------|----------|--------|------------|--------|--------------|--------|
| 2016 | 77,255,731 | | 2,421,880 | | 3014 | |
| 2017 | 85,246,948 | 10.344 | 2,667,223 | 10.130 | 3370 | 11.812 |
| 2018 | 96,311,224 | 12.979 | 2,903,996 | 8.877 | 3747 | 11.187 |
| 2019 | 103,077,736 | 7.026 | 3,088,187 | 6.343 | 3807 | 1.601 |
| 2020 | 103,209,724 | 0.128 | 2,839,977 | −8.037 | 3728 | −2.075 |
| 2021 | 116,619,158 | 12.992 | 3,214,548 | 13.189 | 4209 | 12.902 |
| Average growth | | 8.694 | | 6.100 | | 7.085 |

Source: Compiled and calculated by author from Annual Report 2019–20 and Chittagong Port Authority (CPA) website [30,31].

Some of the other performance indicators including, for example, turnaround time, dwell time, berth occupancy or utilisation, and equipment availability are readily available from the Chittagong Port Authority (CPA) Annual Reports, which are presented in the Table 3.

**Table 3.** Different efficiency indicators.

| Efficiency Indicators | Years | | | | | |
|------------------------|---------|---------|---------|---------|---------|---------|
| | 2014–15 | 2015–16 | 2016–17 | 2017–18 | 2018–19 | 2019–20 |
| Ship turnaround time (days) | 4.26 | 2.79 | 2.83 | 2.68 | 2.88 | 2.86 |
| Dwell time of container (days) | 17.48 | 11.88 | 11.15 | 10.81 | 10.78 | 9.99 |
| Berth occupancy (in %) | 65.04 | 73.95 | 76.93 | 93.38 | 90.55 | 89.42 |
| Equipment availability (in %) | 60.63 | 54.06 | 46.69 | 45.63 | 45.64 | 45.03 |

Source: Compiled by author from CPA Annual Reports [30,32–35].

In Table 3, it is observed that the berth occupancy and equipment availability indicators are showing low performances, and this is because of the lack of efficiency and infrastructure development complying with the growth of container handling. The berth occupancy rate has reached 90%, which is alarming. According to Alderton [36], when the utilisation rate of port resources exceeds 70%, congestion starts. Haralambides [37] pointed out that the utilisation rate is around 75% when congestion starts to set in. On the other hand, turnaround time and dwell time on a year-by-year basis show almost similar durations, which also indicates low performance compared to international practices. At an international level, the median turnaround time is 0.7 days for a container ship and 0.97 days for a typical ship [38]. It is important to mention that the turnaround time calculated by the CPA in Table 3 only has taken into consideration the berthing and unberthing time without incorporating the waiting time. An analysis of the berthing schedule collected from the CPA for 90 vessels in 2020 shows that, before berthing, a container ship needs to wait for 6 days on average (an example berthing schedule is attached as Appendix A). A similar analysis of the berthing schedule of 45 vessels in 2022 shows that ship waiting time on average is 4 days [39]. The port is showing a very high berth occupancy (Table 3), which is an indication of ship congestion in the port. This is a significant delay affecting the supply chain and needs to be considered in the lead time analysis. Based on the above discussion, a process flow of the Chittagong port for the container handling services is

presented in Figure 1, which includes the sequence of different services and the required average time in days. Although Figure 1 only displays the import of containers, the export follows the same process in reverse.

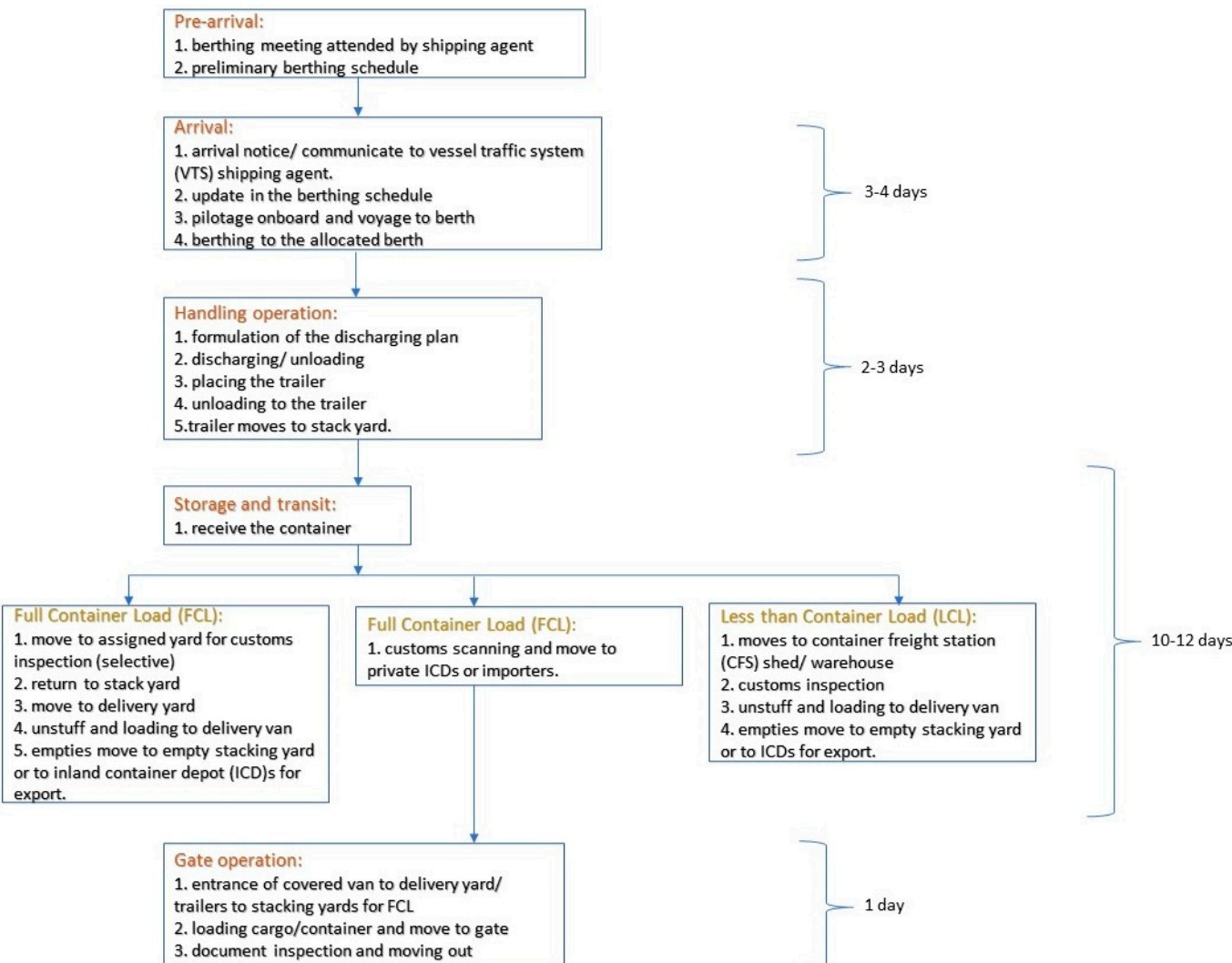

**Figure 1.** The process flow with required times in Chittagong port. Source: prepared by the author based on different sources.

Given the above performance data, the yard utilisation ratio, which is the ratio of the container holding capacity to the number of containers on hand, has been examined for two randomly selected periods of time in June 2020 (an example vessels' particulars and yard utilisation data sheet is attached as Appendix B) and May 2022 [40]. It shows that, in 2020, the yard utilisation ratio was more than 75%, whereas in 2022, it was close to 90%, which is also very high and indicates a container congestion in the yard operation.

The container handling facilities in Chittagong port include the general cargo berth (GCB), the Chittagong Container Terminal (CCT) and New Mooring Container Terminal (NCT). The GCB is traditionally dedicated to the handling of bulk commodities, but to cope with the increasing pressure of the containerised trade, part of it has now been dedicated to container handling. There is a total of 11 berths under the GCB, but 6 berths are used for general cargo handling and 5 berths are used for handling the container ships. In the CCT and NCT, there are 8 berths dedicated to container handling, totalling 13 berths for container handling for the CPA. Table 4 shows the year-wise and terminal-wise infrastructure data.

**Table 4.** Comparative Container Handling Facilities.

| Items | Quantity 2021 | Quantity 2020 | Quantity 2017–18 | Quantity 2015–16 |
|---|---|---|---|---|
| Container berths by number | 13 | 11 | 11 | 11 |
| Holding capacity (GCB + CCT + NCT+) in TEUs | 49,018 | 49,019 | 38,917 | 36,357 |
| Yards in number | 22 | 22 | 22 | 20 |
| Quay gantry crane in number | 14 | 14 | 4 | 4 |
| Rubber tyred gantry crane (40-ton capacity) in number | 41 | 41 | 21 | 19 |
| Rail-mounted yard gantry crane (40-ton capacity) in number | 1 | 1 | - | - |
| Mobile harbour crane (84-ton capacity) in number | 5 | 3 | 2 | 2 |
| Straddle carrier (4 containers high) (40-ton capacity) in number | 44 | 48 | 36 | 38 |
| Straddle carrier (2 containers high) (40-ton capacity) in number | 1 | 2 | 2 | |
| Reach stacker (45-ton capacity) in number | 17 | 11 | 15 | 11 |
| Forklift truck (42-ton capacity) in number | - | 3 | 5 | 5 |
| Forklift (spreader) (16-ton capacity) in number | 4 | 9 | 17 | 19 |
| Reach stacker (7-ton capacity) in number | 9 | 9 | 6 | 6 |
| Container mover (50-ton capacity) in number | - | 8 | 5 | 5 |
| Container mover (32-ton capacity) in number | 8 | - | - | - |

Source: Compiled by author from Chittagong Port Overview [41–44]. Note: the blank cell (-) means there is no such equipment for that particular year.

Among the terminals, the GCB handles all the self-geared vessels. This terminal does not have any quay gantry cranes, whereas CCT and NCT are installed with quay gantry cranes. It is observed that the port had only four quay gantry cranes till the beginning of 2020. The CPA procured 10 new gantries in 2020 leading the ports to currently have 14 gantries in operation. Before these 10 new gantries, the 4 gantries were installed in the CCT. To understand the crane performance, the throughput of the CCT and the NCT is examined. It is found from the CPA officials and operators that the average output of a quay crane is 22 TEUs per hour and an operating hour is around 20 h considering some wastage of time in 2020. An improvement is possible in this area also as, on large ports, the average crane output can reach 25–40 moves per hour [45]. In Australia, it is observed that the operating minutes per moves ranges from 0.92 to 1.32 min [46]. Even if the highest duration of 1.32 is considered, as a general assumption, 45 containers can be handled per hour. According to Bartošek and Marek [47], a quay crane can conduct about 30–50 moves per hour in practice. The type and design of the crane could be a factor here, but, no matter which are used, from the above examples it could be stated that Chittagong port is lagging behind in the context of performance and operational efficiency. In the formulation of different scenarios, all these infrastructural improvement and performance data are taken into consideration to examine the impact of infrastructure and performance improvement for $CO_2$ emissions.

*2.4. Dhaka–Chittagong Hinterland and Its Performance*

The port is around 300 kilometres (km) away from the primary growth centre of the country. The main freight corridor in Bangladesh carries around two million TEUs annually. The corridor is well connected to the Chittagong port via roads, railways and inland waterways [48]. Road transport carries more than 90% of container cargo, while rail transport carries 5–6% and inland waterway transport (IWT) has a share of around 1%. For export and import, the port is supported with a total of 21 inland clearance depots (ICDs). Among these 21 ICDs, 18 are located in the adjacent area of Chittagong port. They are mainly used for stuffing/unstuffing containers. The containers move from the port to the ICDs and are stripped with their contents placed onto trucks or covered vans for inland destinations. In contrast, all the export cargo is moved from their origin, by trucks or covered vans, to the ICDs where it is stuffed in containers to move to the port for final shipment [49]. However, for rail and IWT all the containers are transported to ICDs located in the Dhaka region where the stuffing/unstuffing operations are undertaken. There is one inland container terminal located in Dhaka with an annual handling capacity of 90,000 TEUs, which is connected by rail to the Chittagong port [50]. Two inland container terminals are operating near Dhaka, connected by IWT to Chittagong port, with a total handling capacity of around 250,000 TEUs annually [51,52].

The essential features of the three different modes connecting the Dhaka–Chittagong corridor to the Chittagong port are presented in Table 5. It is observed that the IWT transit time is very long compared to road and rail, and it requires more time due to the lack of regular and timely services. But in the cases of road and rail, there are always problems of congestion.

**Table 5.** Basic features of three different modes.

| Mode | Approximate Average Distance in km | Transit Time in Hours |
|------|-----------------------------------|------------------------|
| Road | 250 | 10 |
| Rail | 310 | 18 |
| IWT | 300 | 36 |

Source: Compilation by the authors based on [53] and data provided by transport operators.

## 3. Methodology and Materials

The fuel consumption in different port activities is very much associated with their time performance. The various units or equipment operating in different areas of ports, including the cargo vessels, service vessels, handling and other terminal equipment, trucks and trailers, are all emitting $CO_2$ emissions, and the longer they operate the more they emit. The concept here is estimating emissions based on energy consumption when using different operational units or equipment to undertake different activities or movements and the distance between activities by different operational units or equipment [3,19,20]. For example, in order to calculate carbon emissions in a container terminal, Budiyanto and Huzaifi [54] categorised the fuel or energy consumed by a container crane, handling equipment and terminal trucks and estimated a certain period of working time. In the case of Chittagong port, a similar methodology is used in which alongside the port activities in CCT and NCT, the emissions from the hinterland distribution for these two terminals are also taken into consideration. After calculating the amount of fuel consumption for different activities, it is then converted to $CO_2$ emissions using the conversion factor from the Fourth IMO GHG study 2020 [55]. Therefore, to calculate the $CO_2$ emissions from CCT and NCT in Chittagong port from shipping, handling, gate operations and the inland distribution including the Dhaka–Chittagong corridor, the following equation, used to keep consistency with Budiyanto and Huzaifi [54], and the operational order discussed in Section 2.3 are used:

$$F_{cph} = F_{cm} + F_{ty} + F_g + F_{cdc} \tag{1}$$

where:

$F_{cph}$ ~ fuel consumption in port and hinterland;
$F_{cm}$ ~ fuel consumption from container ship and marine operation;
$F_{ty}$ ~ fuel consumption in terminal and yard;
$F_g$ ~ fuel consumption in gate operation;
$F_{cdc}$ ~ fuel consumption in Dhaka–Chittagong corridor.

Looking at the performance data presented in Section 2 (Sections 2.3 and 2.4), it is observed that there are plenty of opportunities to improve the performance in the port as well as the hinterland area. In the case of marine operations, the ship waiting time and turnaround time were considered for, respectively, 3 and 2 days in 2020 (Figure 1), during which the ships were generating $CO_2$ emissions as the auxiliary engines of the ships were running. Therefore, the improvement in the ship waiting time and turnaround time by reducing the current time of 5 days can achieve $CO_2$ emission reduction. It is notable that the reduction in the turnaround time will enable the port to handle more containers, which requires more tugging and other services and may increase $CO_2$ emissions, but the per TEU $CO_2$ emissions might be reduced. In the case of the terminal and yard operations, the performance measurement is considered from two aspects: one aspect is the number of days the container is staying at the yard, and the other aspect, concerning productivity, is the operation hours and the per hour output of different equipment. Taking into consideration the above two aspects, this will also result in higher $CO_2$ emissions due to the increase in employment hours of the equipment as a result of performance improvement. Although this is in the case of overall emissions only, the per TEU emissions might be reduced. In the case of gate operations, the duration of trucks in the port and the extra combustion of fuel due to congestion are measured. All these data are discussed in Section 2.3 and collected from the CPA web sources, and some missing data from the Port and Traffic Department of the CPA are included in the relevant sections of the paper.

In the case of the hinterland, as roads carry more than 90% of containers sent to the Dhaka region, and since the empirical research suggests the possibility of a modal shift [53], the saving of fuel due to a shift of containers to a more economically and environmentally efficient mode of transport from the road is calculated. In calculating the $CO_2$ emissions for three different modes connecting the Dhaka–Chittagong hinterland, the data on the types of vehicles and the transport and other infrastructural facilities are important to use, at least, for road transport as road transport comprises more than 90% of the market share for containers. In order to understand emissions from road transport on this corridor, the various road transport factors along the corridor, including the types of vehicles, average speed, distance and the fuel efficiency data, are collected from the draft Road User Charge 2016–17 report published by the Roads and Highways Department, Bangladesh [56]. According to this report [56], the freight transport in Bangladesh is composed of three types of trucks, namely, heavy and articulated trucks, medium trucks and small trucks. It defines the heavy and articulated trucks as those that can carry more than 7.5 tons of payloads. The medium trucks are those that can carry 3 tons of payloads. The registered trucks operating in the country show that 2% of trucks are articulated, 47% are heavy trucks and 16% are medium trucks. Therefore, considering the loading capacity of these trucks, it seems that only the heavy and articulated trucks are the most suitable carriers to run in the Dhaka–Chittagong corridor. However, it is observed that some medium trucks are also involved in carrying goods along the corridor [57]. Therefore, in calculating the emissions, mostly heavy trucks in operation are considered, while the operation of very few articulated and medium trucks are considered for average fuel consumption data. As a result, the following equation from Uddin and Mizunoya [57] is used, which includes activity of the truck such as the total amount of traffic, travel time, average speed and the fuel consumption rate:

$$F_{cdc} = N_{traffic} * T_t * \frac{v}{F_e} \tag{2}$$

where:

$F_{cdc}$~fuel consumption in Dhaka–Chittagong hinterland;

$N_{traffic}$~number of vehicles in traffic;
$T_t$~travel time;
$v$~average speed;
$F_e$~fuel efficiency.

## 4. CO$_2$ Emissions in Chittagong Port and Dhaka–Chittagong Hinterland

*4.1. CO$_2$ Emissions and Findings from the Simulation-Based Scenario Implementation in Chittagong Port*

Table 6 presents the estimated CO$_2$ emissions for Chittagong port only. The hinterland part is calculated in a later section, and the two are finally combined to understand the total port-hinterland emissions.

**Table 6.** Estimated CO$_2$ emissions in 2020 based on the number of equipment and ship calls.

| Emissions from Different Activities | Working Time in Days/Hours | Fuel Consumption in Litre/Days or Hours | No. of Ships | Total Fuel Consumption in Tons | Emissions in Tons | Per TEU Emissions in kg |
|---|---|---|---|---|---|---|
| Ships and marine operation | | | | | | |
| Container ships | 5 days on average, only auxiliary engine is running | 2000 | 1050 | 10,500.00 | 33,663.00 | 19.80 |
| Service vessels—Tug | 2 h | 576 | 1050 | 1209.60 | 3799.35 | 2.23 |
| Service vessels—others | 0.5 h | 155 | 1050 | 81.38 | 255.60 | 0.15 |
| Sub-total | | | | 11,790.98 | 37,717.95 | 22.19 |
| Handling and yard operation | | | | | | |
| CPA | | | | | | |
| RTG | 20 h | 23 | 41 | 4356.66 | 13,684.27 | 8.05 |
| SC | 20 h | 18 [1] | 48 [2] | 3991.68 | 12,537.87 | 7.38 |
| RST-45 tons | 20 h | 15 | 11 | 762.30 | 2394.38 | 1.41 |
| RST-7 tons | 20 h | 13 | 9 | 540.54 | 1697.84 | 1.00 |
| FLT-16 tons | 20 h | 13 | 9 | 540.54 | 1697.84 | 1.00 |
| Sub-total | | | | 10,191.72 | 32,012.19 | 18.83 |
| Private operator's equipment, Sub-total [3] | | | | 4184.95 | 13,144.93 | 7.73 |
| Sub-total | | | | | | 26.56 |
| Gate operation | 4 h | 4 | 3000 | 17,640.00 | 55,407.24 | 32.59 |
| Total | | | | 43,807.65 | 138,282.31 | 81.34 |

Source: Calculated by author using data and information from CPA performance data [31,58], annual report [30,35] and terminal operators. [1] Consumption per hour for some equipment matches results from [59]. [2] A total of 60% use is taken into account, here, considering the CCT and NCT handle 60% of the containers. [3] This data is from the private operator, who has only provided the average total fuel consumption data.

In the above calculation, the fuel consumption only considers an average size of vessel carrying 2000 TEUs, which is the average size of container vessels entering the port (the average size of ships and average container carrying capacity for the year 2020 is, respectively, considered 19,300 GRT and 1722 TEUs according to [60]). It is important to mention that the consumption of the main engine within port movement is not taken into consideration for conducting the simulation. This is because no scenarios for the improvement of vessel speed in the channel is considered. Taking into account the channel

characteristics in terms of depth, width and sharp bends, it does not seem possible to achieve a considerable improvement in terms of speed increase/decrease within this channel. Therefore, the vessel movement within the port from anchorage to berth will not change even if the performance in waiting and berthing is improved. As a result, the consumption of the auxiliary engine is only considered in the simulation rather than the main engine. It is observed from Table 5 that the total fuel consumption and $CO_2$ emissions from the container vessels entering the CCT and NCT in 2020 are 10,500 tons and 33,663.00 tons, respectively. The fuel consumption per vessel is 10,000 kg and the $CO_2$ emissions per TEU are 19.80 kg from container vessels. The total per TEU emissions for the overall marine operations including the allied tug and pilotage services is 22.19 kg per TEU. For terminal handling, $CO_2$ emissions are 26.56 kg per TEU, whereas, in the gate, they are 32.59 kg per TEU. The total emissions including the waiting, berthing, terminal operation and final delivery through the gate are 81.34 kg $CO_2$ per TEU. There are not many similar studies in the literature, except for Huzaifi and Budiyanto [12], which found that carbon emissions in the Jakarta International Container Terminal are 29.08 kg per TEU, but, as mentioned earlier, their study considered the inward services from berthing to taking the containers out of port. Another study by Yang et al. calculated that the $CO_2$ emissions per TEU in the port of Shenzhen are 23.49 kg [22,23]. Comparing the fuel consumption with previous empirical findings, noticeable differences can be seen, which are mainly due to the performance gap between Chittagong port and the above ports. That is, the performance of Chittagong port in every aspect of waiting, berthing, dwell time and gate operation time is not up to the mark. Therefore, policymakers have the opportunity from this calculation to look at the emissions status of the port and the performance issue from an environmental sustainability perspective.

The results from the simulation for 2020, 2022 and different scenarios are presented in Table 7. As highlighted in the literature review section, the formulation of the scenarios is based on the infrastructural development and performance and operational efficiency of the CPA. As we can see, some infrastructural improvement was conducted from 2020 to 2022 (Table 4); therefore, the impact of this improvement has been studied in the simulation. Moreover, due to the improvement in the infrastructure, changes in the performance and operational efficiency, mostly indicating improvement in the working hours and handling rate of gantries, are considered. As discussed in Section 2.3, the working hours for gantries were 20 h, and the handling rate was 22 TEUs per hour in 2020; therefore, some improvement in the performance of these indicators is considered. Moreover, due to the improvement in the handling of gantry cranes along with other associated equipment, an improvement in the waiting and turnaround times is also considered. As discussed before, the median turnaround time for container ships is 0.7 days [38] in ports, and on large ports, the average crane output can reach 25–50 moves per hour [45–47]. Therefore, the scenarios include an increase in the working hour of cranes and equipment from 20 h to 22 h; an increase in the handling rate of gantries and other equipment from 22 TEUs per hour to 24 TEUs per hour; and, finally, an impact of the handling operation on the yard handling time reduction and a reduction in the customs time, which will have a corresponding impact on the waiting and turnaround time. It is important to discuss that, in the simulation analysis, the waiting time of ships, berthing duration of ships, number of ships in the berth, duration of a container staying at the yard, number of quay cranes per ship and their output, equipment availability and daily containers handling in the yard, and trucking and gate operations are considered as the input variables. Among these input variables, some are considered random, and some are considered fixed. The random input variables and their probability distributions are shown in Appendix C. As discussed, the data for all these input variables are collected from the CPA web sources and some missing data from the Port and Traffic Department of CPA, examples of which are attached in Appendices A and B. On the other hand, the total lead time for containers, the throughput, the fuel consumption in various operations and the $CO_2$ emissions are considered the output variables in the simulation analysis. A sample sheet of the simulation is presented in Appendix C.

**Table 7.** Simulation and scenarios findings.

| Sl. No. | Emissions from Different Activities | Average of Total and per TEU Emissions | | | |
|---------|-----------------------------------|------|---------------|---------|---------|
| | | **Mean** | **St. Deviation** | **Maximum** | **Minimum** |
| | | In 2020 | | | |
| | | Total emissions in tons | | | |
| 1. | Total from CCT and NCT | 150,310.44 | 20,042.07 | 184,646.70 | 116,759.65 |
| | | Per TEU emissions in kg | | | |
| 2. | Ship and marine operations | 35.09 | 13.30 | 61.53 | 13.92 |
| 3. | Handling and yard operations | 24.86 | 3.09 | 36.34 | 21.69 |
| 4. | Gate operations | 31.96 | 3.97 | 46.71 | 27.89 |
| 5. | Per TEU from CCT and NCT | 91.92 | 17.39 | 141.32 | 65.39 |
| | | In 2022 | | | |
| | | Total emissions in tons | | | |
| 1. | Total from CCT and NCT | 151,366.79 | 19,429.13 | 186,731.64 | 116,600.39 |
| | | Per TEU emissions in kg | | | |
| 2. | Ship and marine operations | 32.26 | 12.18 | 60.24 | 12.09 |
| 3. | Handling and yard operations | 23.52 | 2.96 | 33.03 | 20.47 |
| 4. | Gate operations | 31.75 | 4.00 | 44.59 | 27.63 |
| 5. | Per TEU from CCT and NCT | 87.52 | 16.11 | 134.26 | 63.71 |
| | | Scenario I: Increase in quay crane working hours | | | |
| | | Total emissions in tons | | | |
| 1. | Total from CCT and NCT | 152,313.51 | 19,523.20 | 188,557.34 | 119,548.19 |
| | | Per TEU emissions in kg | | | |
| 2. | Ship and marine operations | 27.29 | 10.87 | 51.40 | 10.07 |
| 3. | Handling and yard operations | 23.16 | 2.43 | 31.44 | 20.47 |
| 4. | Gate operations | 28.42 | 2.99 | 38.57 | 25.12 |
| 5. | Per TEU from CCT and NCT | 78.87 | 13.77 | 117.24 | 57.90 |
| | | Scenario II: Increase in quay crane per hour TEU handling rate | | | |
| | | Total emissions in tons | | | |
| 1. | Total from CCT and NCT | 152,059.51 | 19,938.91 | 186,713.69 | 117,143.49 |
| | | Per TEU emissions in kg | | | |
| 2. | Ship and marine operations | 24.25 | 10.21 | 44.06 | 6.88 |
| 3. | Handling and yard operations | 23.10 | 2.58 | 29.88 | 20.53 |
| 4. | Gate operations | 26.76 | 2.99 | 34.62 | 23.78 |
| 5. | Per TEU from CCT and NCT | 74.11 | 13.10 | 105.95 | 53.38 |
| | | Scenario III: Reduction in yard operation and customs time | | | |
| | | Total emissions in tons | | | |
| 1. | Total from CCT and NCT | 147,338.87 | 18,868.48 | 179,019.49 | 113,937.69 |
| | | Per TEU emissions in kg | | | |
| 2. | Ship and marine operations | 23.52 | 10.09 | 44.41 | 6.65 |
| 3. | Handling and yard operations | 21.99 | 2.56 | 29.59 | 19.38 |
| 4. | Gate operations | 26.98 | 3.15 | 36.31 | 23.78 |
| 5. | Per TEU from CCT and NCT | 72.49 | 13.48 | 109.68 | 52.43 |

Source: Calculated by author.

The simulation results show that via the implementation of different scenarios the per TEU $CO_2$ emissions can be reduced from 91.92 kg in 2020 to 72.49 kg under Scenario III, that is, a reduction in the yard operation and customs time (duration), as this will impact the waiting and turnaround time. A step-by-step implementation of the different scenarios and the respective findings are discussed in the following parts. From Table 7, it is observed that, even though there are some fluctuations in the total emissions, the mean emissions per TEU reduce from scenario to scenario, and there are further emission reduction opportunities with the existing capacity if a desired performance is achieved, which is examined in the simulation.

In 2020, the average total emissions were 150,310.44 tons, and they could be as low as 116,759.65 tons. The emissions per TEU were 91.92 kg on average, but could be as low as 65.39 kg. In 2022, the average total emissions rose a bit to 151,366.79 tons, but the emissions per TEU reduced to 87.52 kg on average and could be as low as 63.71 kg per TEU. In 2022, the operation of additional gantry cranes and other allied equipment were taken into consideration in the port operations as the port was supplied with new gantry cranes and other equipment starting from 2020 (Section 2.3 and Table 4). The operation of additional equipment has increased the total emissions, but the emissions per TEU show a reduction.

(i) Scenario I—Increase in the working hours for the quay crane

Under Scenario I, the working hour of the equipment increases from the base 20 h to 22 h, and the total emissions increase to 152,313.51 tons on average and the emissions per TEU decrease to 78.87 kg on average and could be as low as 57.90 kg. The increase in the total emissions seems not that high even though all the equipment is operating longer compared to 2022. This could be mainly due to the reason that a performance improvement of the handling operation always results in the reduction in the ships waiting and berthing time. This has reduced the total emissions to some extent, which has traded-off the increase in total emissions due to the equipment operational time increment.

(ii) Scenario II—Increase in the TEU handling rate per hour for the quay crane

In the case of Scenario II, a decrease in the total emissions is observed and the emissions per TEU decrease further to 74.11 kg and could be as low as 53.38 kg on average. In this case, an increased output per hour for gantries and other equipment is considered. In the base case, the output of gantries is 22 TEUs per hour, which increases to 24 TEUs per hour under this scenario. This increased the total throughput without impacting the consumption of fuel as the gantries and other equipment became more efficient and were able to handle comparatively more containers while their operating duration remained the same. Moreover, it improved performance impacts on the berthing and waiting time for ships. As a result, the total emissions as well as the emissions per TEU are reduced.

(iii) Scenario III—Port reduces the yard operating time and a simultaneous reduction in customs services

Under Scenario III, a reduction in the time required in the yard and for the customs procedures is considered. It is observed from Figure 1 that the port requires a container to stay in the yard for 10–12 days on average for customs and other inspections and, finally, to unstuff and load the contents in a trailer, truck or covered van for final delivery. Moreover, the improvement in the quay operation in scenario II requires a simultaneous operational improvement in the yard as well; otherwise, there will be congestion in the yard leading towards a deterioration of achievement in the quay operation. Therefore, in the third scenario an improvement in the yard operation (duration) and customs services is considered to understand the overall impact on the emissions. This improvement has further impacted the waiting and berthing times without any increase in the equipment operation hours. The same number of containers are handled by the yard and customs but with higher efficiency in doing the same work. This impacts the total as well as the emissions per TEU. The average total emissions reduce to 147,338.87 tons, and the emissions per TEU reduce to 72.49 kg and could be as low as 52.43 kg on average.

Via the implementation of these scenarios, it is observed that the Chittagong port can reduce around 20 kg $CO_2$ emissions per TEU on average. Considering that a total handling in the CCT and NCT is 1.7 million TEUs, a total of 34,000 tons of $CO_2$ emissions could be reduced. If we consider the lowest possible $CO_2$ emissions, the reduction in $CO_2$ emissions per TEU could be as low as 40 kg of $CO_2$ emissions, totalling to 68,000 tons of $CO_2$ emissions that could be reduced in the CCT and NCT. If this reduction in $CO_2$ emissions is considered for the total throughput of Chittagong port, which is 3 million TEUs, the country can save a total of 60,000 tons of $CO_2$ emissions from Chittagong port operation on average, which could be as low as 120,000 tons of $CO_2$ emission reduction.

### 4.2. Calculation of $CO_2$ Emissions in the Dhaka–Chittagong Hinterland and the Findings from the Modal Shift

In order to understand the fuel efficiency, certain information on the types of trucks are needed, which is also collected from the Road User Charge 2016–17 report [56]. In Table 8, the particulars of the trucks are provided. The fuel efficiency for these types of trucks is either 4–6 km/L, or 17–25 L/100 km.

**Table 8.** Truck particulars.

| Motorised Vehicle | Description | Most Popular Brand | Model | CC | Cylinders | HP | Tires |
|---|---|---|---|---|---|---|---|
| Articulated and heavy truck | Three or more axles. Includes multi-axle tandem trucks, container carriers and other articulated vehicles. | Tata (44%) | Tata LPT 2516 | 5883 | 6 | 157 | 10 |
| Medium truck | Two axle rigid trucks. >three tonne payload agricultural tractors and trailers are also included | Tata (23%) | Tata LPT 1615 | 5883 | 6 | 145 | 6 |

Source: Compiled from Road User Charge 2016–17 report [56].

The average speed of trucks on the highway is 30 km/h [56]. In the case of the Dhaka–Chittagong corridor, the average time is considered to be 10 h and the distance is 250 km. For rail, the distance is 310 km, and, for IWT, it is 300 km (Table 5). However, for both the rail and IWT, around 25 km of last/first mileage trucking needs to be considered to calculate $CO_2$ emissions. Chittagong port handled 2.7 million TEUs and 25.7 million containerised tons in the 2020–21 financial year. As discussed, 70% of this is destined for and originated from Dhaka and the Dhaka–Chittagong corridor is carrying this volume. Road is carrying more than 90%, rail is carrying 5–6% and IWT is carrying 1%. Therefore, road is carrying around 16.7 million tons or 1.8 million TEUs. As a result, it is assumed that at least 2.5 million truck trips in both directions are required. Consequently, considering Equation (2), the fuel consumption and the same conversion factor from the Fourth IMO GHG study [55], the emissions figure for road transport is presented in Table 9. In the cases of rail and IWT, it is not possible to calculate the emissions using Equation (2) due to the poor data availability, which can be overcome by using the emissions factor from McKinnon and Piecyk [28], the IPCC [25] and Wang and Peng [9]. An indicative emissions factor for all the three modes based on these three studies are proposed in Section 2.2, which is 180 $gCO_2$/ton-km for road, 26 $gCO_2$/ton-km for rail and 34 $gCO_2$/ton-km for IWT. Moreover, as mentioned earlier, the ranges in the emissions data for all studies show a variety of emissions depending on the vehicle, fuel, infrastructure and even the behaviour of the drivers. Considering the socio-economic context of the hinterland, the highest possible emissions factor is considered while calculating the emissions per TEU for rail and IWT. In order to have a comparative picture, in addition to the emissions calculation of road transport using Equation (2), emissions are also calculated using the $gCO_2$/ton-km

emissions factor proposed by McKinnon and Piecyk [28], the IPCC [25] and Wang and Peng [9] and presented in Table 9. Since the McKinnon and Piecyk [28] and the IPCC [25] reports produced the emissions factor in $gCO_2$/ton-km, the total TEU tonnage is therefore divided by the total number of TEU and used to convert the $gCO_2$/ton-km to $gCO_2$/TEU-km and finally multiplied by the distance and divided by 1000 to convert it to $CO_2$ emissions in kg/TEU. A comparison of $CO_2$ emissions per TEU for the Dhaka–Chittagong corridor is presented in Table 9 below.

**Table 9.** $CO_2$ emissions for different modes in the Dhaka–Chittagong corridor.

| Transport Mode | Equation (2) | An Average from Table 7 |
|---|---|---|
| | $CO_2$ Emissions kg/TEU | $CO_2$ Emissions kg/TEU |
| Road | 363.01 | 417.94 |
| Rail–Road | | 107.72 |
| IWT–Road | | 128.24 |

Source: Calculated by the author.

From the above table, if we consider that the $CO_2$ emissions for road are 363.01 kg/TEU and the emissions for rail–road and IWT–road are, respectively, 107.72 kg/TEU and 128.24 kg/TEU, it seems that rail emits the least $CO_2$ for modes of carrying containers in the Dhaka–Chittagong hinterland. It is important to mention that, although it was not possible to calculate the approximate emissions per TEU for rail and IWT using the real scenarios for the hinterland, among the global emissions data ranges as proposed by McKinnon and Piecyk [28] and the IPCC [25], the highest emissions data for these two modes were used to estimate the emissions per TEU. Therefore, the calculated differences of emissions among different modes specifically between road vs. rail and road vs. IWT will be at least an indicative measurement and could be close or lower than the actual or real emissions. Therefore, it can at least provide the minimum of $CO_2$ savings for modal shifting. A modal shift of containers from road to rail can reduce per TEU emissions by 255.29 kg and to IWT per TEU by 234.76 kg. Considering the total TEUs using the Dhaka–Chittagong hinterland, which is 1.8 million TEUs, if 50%, or 900,000 TEUs, can be shifted to rail or IWT, approximately a total of 229,500 tons or 211,000 tons of $CO_2$ emissions can be reduced. Moreover, a modal shift of 900,000 TEUs to rail or IWT means that the current number of truck entrances to Chittagong port will reduce by a substantial number, meaning a further reduction in the port emissions by around 29,333.25 tons of $CO_2$ emissions.

## 5. Summarising the Findings

By implementing different scenarios using a simulation technique, this paper first finds that the significant reduction in $CO_2$ emissions can be achieved in the case port. In calculating the $CO_2$ emissions, activity-based formulae have been used where the fuel or energy consumption from the marine operation and container ships, terminal and yard operations and gate operations are being summed up. To calculate the emissions from different operations, the working hours of different equipment and their per hour fuel consumption has been considered, and finally, using the Fourth IMO GHG study 2020 [55] conversion factor, the total as well as the emissions per TEU in ports are calculated. From the different scenarios of the simulation, the average emissions per TEU in the CCT and NCT in 2020 were 91.92 kg but could be as low as 65.71 kg. In 2022, the emissions per TEU on average reduced to 87.52 kg and the minimum was 63.71 kg. Via the implementation of the other scenarios, such as increasing the equipment working hours, increasing the handling output of gantries per hour and reducing yard operation time and customs time, it was demonstrated that the average emissions per TEU can be, respectively, 78.87 kg, 74.11 kg and 72.49 kg, and these can be as low as 57.90 kg, 53.38 kg and 52.43 kg respectively. It is observed that, in this way, the Chittagong port can reduce around 20 kg of $CO_2$ emissions per TEU operation on average, which could be as low as 40 kg of $CO_2$ emissions per TEU.

Considering the total handling in the CCT and NCT, which is 1.7 million TEUs, a total of 34,000 tons of $CO_2$ emissions could be reduced, and the lowest possible $CO_2$ emissions could be of 68,000 tons of $CO_2$ emissions. If this reduction is considered for the total throughput of Chittagong port, which is 3 million TEUs, the country can reduce a total of 60,000 tons of $CO_2$ emissions from Chittagong port operations on average and 120,000 tons of $CO_2$ emission reduction in the lowest possible emissions case.

Furthermore, the emissions from the hinterland are also analysed from the perspective of modal shift to low carbon emissions modes among the four widely discussed strategies, which include (i) avoiding journies when possible, (ii) a modal shift to a lower-carbon transport mode, (iii) lowering energy intensity and (iv) reducing the carbon intensity of fuel. This shows that shifting to rail or IWT from road can reduce $CO_2$ emissions by 255.30 kg and 234.75 kg per TEU, respectively. Considering the total TEUs using the Dhaka–Chittagong hinterland, which is 1.8 million, if 50% of that can be shifted to rail or IWT from a policy or other operational and infrastructural perspectives, approximately a total of 229,500 tons or 211,000 tons of $CO_2$ emissions can be reduced. Moreover, a modal shift of 900,000 TEUs to rail or IWT will reduce the entrance of the current number of trucks entering the Chittagong port, meaning a reduction in the port emissions by around 29,333.25 tons of $CO_2$ emissions. It is important to mention that, in the calculation of the emissions from road, the data from the Road User Charge 2016–17 report from the Bangladesh Roads and Highway department has been used, which includes the particulars and fuel efficiency of the types of vehicles operating in the Dhaka–Chittagong corridor. But in the cases of rail and IWT, the emission figures have been calculated based on the various global secondary studies as the vehicle particulars and fuel efficiency data for these two modes are not readily available.

To sum up the above findings, it can be argued that the overall $CO_2$ emission reduction for one TEU to travel from Dhaka region through the Chittagong port using rail or IWT instead of road transport could be from 454.93 kg to 170.90 or 190.90 kg on average and could be as low as 150.40 or 170.45 kg. Considering the total emission reduction, if 900,000 TEUs can be shifted to rail or IWT, the country can reduce a total of 319,091.20 or 300,620.95 tons of $CO_2$ emissions, which could be as much as 379,091.20 or 360,620.94 tons of $CO_2$ emissions, which would be a very important achievement for the overall transportation system.

## 6. Conclusions

This paper shows that by implementing different scenarios for infrastructure development and performance and operational efficiency improvement, the $CO_2$ emissions in ports can be reduced significantly. Moreover, the modal shift from road to rail or IWT can also create significant emission reduction for container transportation from Dhaka using the Chittagong port for international shipment. The scenarios for the port infrastructure and operational efficiency include the inclusion of new cranes and other equipment, an increase in operation time, an increase in per hour handling quantity and a reduction in yard operation and customs operation time. All these scenarios comply with most of the scenarios that are observed in the literature review section. Furthermore, the modal shift from road to rail or IWT, which is one of the methods for achieving sustainability in the transport sector, is implemented in this research. As discussed in Section 3, there is also empirical evidence that the modal shift to rail or IWT is very much possible for the hinterland. Therefore, the scenarios developed and implemented in this research can contribute considerably to the sustainable port operation for the case port and its hinterland.

For the case port-hinterland, this research first demonstrates the possibility of sustainable port-hinterland transport. It further indicates that the ports in developing countries like Bangladesh have a great scope to concentrate on operational efficiency as well as infrastructure development to achieve sustainability in the port-hinterland transport system. At the same time, this research could benefit researchers as well as practitioners, explaining the implication of operational performance improvement and infrastructure development on the sustainable port-hinterland container transport system. The port operators, port authorities and policy makers can focus on these areas for $CO_2$ emission reduction. A policy could focus on the maximization of resource utilization and the addition of equipment in the short term and could gradually invest in additional infrastructure. A reduction in $CO_2$ emissions from such initiatives can contribute to the air quality and liveability in the port-hinterland neighbourhood as well as the port–city interaction and relationship.

Scenarios such as increasing/decreasing speed in the channel and truck entrance optimization are excluded in this paper. Considering the geophysical characteristics of the channel, including the length, width and bends in several places as well as the current traffic, changes in the speed of vessels in the channel might not be possible. On the other hand, the research is emphasising the modal shift from road to rail or IWT, which will reduce the number of trucks entering the port. Via the implementation of the modal shift, there will be already fewer trucks and less congestion, which will improve the current combustion scenarios and improve the reduction in the emissions from gate operations improving the queuing of trucks. One potential limitation for the emissions calculation was poor data availability (e.g., fuel type, fuel efficiency and the breakup of energy sources) for rail and IWT transportation to calculate the fuel consumptions. As a result, in calculating the emissions for these two modes, some of the examples from the literature are used. This might not be providing the most accurate picture of emissions, but for both rail transport and IWT, the highest possible emissions are taken into consideration so that the indicative reduction in $CO_2$ emissions could provide the minimum possible reduction for the studied hinterland. Further research including the primary investigation and data collection for the port-hinterland operation will provide a much clear picture for the case port.

**Author Contributions:** K.R.H.: Conceptualisation, methodology, data curation, analysis and writing—original draft preparation. W.Z.: Writing—review and editing and supervision. W.S.: Writing—review and editing and Supervision. All authors have read and agreed to the published version of the manuscript.

**Funding:** This research received no external funding.

**Institutional Review Board Statement:** Not applicable.

**Informed Consent Statement:** Not applicable.

**Data Availability Statement:** The performance data can be found at different annual reports from http://www.cpa.gov.bd accessed on 20 May 2022. Fuel consumption data were collected personally from the Chittagong port, authority officials and private operators and are discussed in the relevant section of the paper.

**Conflicts of Interest:** The authors declare that there is no conflict of interest regarding the publication of this paper.

# Appendix A

| PERMISSIBLE F.W.DRAUGHT: | BERTHING POSITION & PERFORMANCE OF VESSELS: DATED : 14/06/2020: | DATED: 15/06/2020, 16/06/2020, 17/06/2020, 18/06/2020, 19/06/2020 & 20/06/2020: |
|---|---|---|

DATED : 15/06,16/06,17/06, 18/06,19/06 & 20/06/20
INWARD: 9.50 9.50 9.50 9.50 9.50 9.50
OUTWARD: 9.50 9.50 9.50 9.50 9.50 9.50

www.cpa.gov.bd

| | TIME HGT: | TIME HGT: | TIME HGT: | TIME HGT: | TIME HGT: | TIME HGT: |
|---|---|---|---|---|---|---|
| L.W.: | 0251 1.15 | 0355 1.12 | 0454 1.01 | 0544 0.89 | | |
| H.W.: | 0906 3.71 | 1010 3.89 | 1101 4.11 | 1143 4.30 | 0002 4.00 | 0039 4.09 |
| L.W.: | 1532 1.31 | 1638 1.18 | 1735 1.03 | 1823 0.92 | 0626 0.80 | 0704 0.75 |
| H.W.: | 2125 3.64 | 2228 3.73 | 2320 3.87 | | 1219 4.42 | 1251 4.50 |
| L.W.: | | | | | 1903 0.88 | 1941 0.87 |

| JETTY NO. | NAME OF VESSEL | LENGTH | CARGO | LAST PORT | FLAG | LOCAL AGENT | DT. OF ARRIVAL | BERT-HING | SHIF-TING | LEA-VING | IMPORT DISCH. | B.ON BOARD | EXP: LIFTED | T.ON BOARD | NAME OF VESSELS |
|---|---|---|---|---|---|---|---|---|---|---|---|---|---|---|---|
| J/2 | XIN HAI ZHOU 3(IMO:9444182) | 139.13 | GI(SCRAP) | BUSAN | PANA | SEAWORLD | 10/06 | 13/06 | - | - | 400 | 9800 | | | |
| J/3 | *NORDTRUST(IMO:9454187) | 189.99 | GI(ST.COIL) | SING | PANA | SPECTRUM | 02/06 | 10/06 | - | 14/06 | 6098 | X | | | SPRING AMIR |
| J/4 | (NO SPACE) | | | | | | | | | | | | | | |
| J/5 | KONG QUE SONG(IMO:9416769) | 179.50 | GI | YANG | HONG | COSCOL | 11/06 | 13/06 | - | - | 304 | 2562 | | | |
| J/6 | MARIA THEO1(IMO:9384863) | 169.26 | LOG | KAND | PANA | RENU | R/A | 11/06 | - | - | 676 | 9838 | | | |
| J/7 | OCEAN FUTURE(IMO:9418779) | 190.00 | GI(SCRAP) | SING | MALT | SRSL | 05/06 | 12/06 | - | - | 2115 | 26360 | | | |
| J/8 | CHARLENE(IMO:9125360) | 170.00 | GI(SCRAP) | KINU | TUVA | EVERETT | 06/06 | 12/06 | - | - | 3065 | 14710 | | | |
| J/9 | ALENTEJO(IMO:9626118) | 179.99 | GI(ST.COIL) | INDO | MARSH | LITMOND | 08/06 | 12/06 | - | - | 5590 | 16181 | | | |
| J/10 | ELA(IMO:9516777) | 175.50 | CONT | P.KEL | SING | KARNAPHULI | 08/06 | 12/06 | - | - | 251 | X | 525/ | | |
| J/11 | MAERSK BINTULU(IMO:9840702) | 185.99 | CONT | S.HAI | SING | MBDL | 06/06 | 11/06 | - | - | 422 | 119 | 666/ | | |
| J/12 | (NO SPACE) | | | | | | | | | | | | | | |
| J/13 | MOUNT KELLETT(IMO:9760627) | 169.99 | CONT | SING | HONG | APL | 11/06 | 13/06 | - | - | 686 | 507 | | | |
| CCT/1 | *XPRESS KOHIMA(IMO:9155016) | 168.00 | CONT | COL | TUVA | SEACON | 03/06 | 11/06 | - | - | 6 | X | 802/ | | OEL DELTA |
| CCT/2 | | | | | | | | | | | | | | | |
| CCT/3 | THORSKY(IMO:9162277) | 183.65 | CONT | COL | CYP | MSCL | 11/06 | 14/06 | - | - | - | 1169 | | | |
| NCT-1 | | | | | | | | | | | | | | | |
| NCT-2 | CNC NEPTUNE(IMO:9836658)(GL) | 172.00 | CONT | SING | SING | APL | 07/06 | 13/06 | - | - | 568 | 604 | 194/ | | |
| NCT-3 | WARNOW MATE(IMO:9509786)(GL) | 180.36 | CONT | P.KEL | CYP | TML | 10/06 | 13/06 | - | - | 879 | 22 | 209/ | | |
| NCT-4 | XPRESS KABRU(IMO:9793727)(GL) | 186.00 | CONT | SING | SING | SEACON | 05/06 | 11/06 | - | - | 123 | X | 586/ | | |
| NCT-5 | *ORNELLA(IMO:9534660) | 172.00 | CONT | SING | PORTU | CT | 04/06 | 11/06 | - | - | 86 | X | 785/ | | KOTA RUKUN |

| CCJ | | | | | | | | | | | | | | | |
|---|---|---|---|---|---|---|---|---|---|---|---|---|---|---|---|

| GSJ | | | | | | | | | | | | | | | |
|---|---|---|---|---|---|---|---|---|---|---|---|---|---|---|---|
| TSP | | | | | | | | | | | | | | | |
| DOJ-3 | REEM 1(IMO:9687631) | 102.51 | BITUMEN | HAMAR | PANA | GOSL | 14/06 | 14/06 | - | - | | | | | |
| DOJ-4 | *TMN PRIDE(IMO:9333254) | 179.99 | HSFO | SING | THAI | AQUAT | 11/06 | 11/06 | - | 14/06 | | | | | |
| DOJ-5 | *NAVIG8 AMETHYST(IMO:9714501) | 184.60 | GAS OL | TANJ | MARSH | PRIDE | 12/05 | 12/06 | - | 14/06 | | | | | |
| DOJ-6 | | | | | | | | | | | | | | | B.SHOURABH |
| DOJ-7 | | | | | | | | | | | | | | | |
| DD | | | | | | | | | | | | | | | |
| DDJ-1 | OCEAN LOVE(IMO:9641352) | 189.99 | STONE | KAND | LIBE | VISION | 07/06 | 10/06 | - | - | | | | | |
| DDJ-2 | | | | | | | | | | | | | | | |
| RM/8 | | | | | | | | | | | | | | | |
| RM/9 | TUG JIAO GONG 83 / TUG QIMAOTIONG 11 HAO / BARGE DA ZHUANG | 37.70 / 20.00 / 50.00 | BALLAST / BALLAST / BALLAST | PAYRA / PAYRA / PAYRA | CHINA / CHINA / CHINA | ??? / ??? / ??? | 04/01 / 04/01 / 04/01 | 04/01 / 04/01 / 04/01 | - | - | | | | | |
| RM-10 | *AU LIBRA(IMO:9236339) | 124.00 | RBD | PADAN | PANA | MTCL | 09/06 | 12/06 | - | - | | | | | |

**Figure A1.** Copy of Berthing Schedule.

**Appendix B**

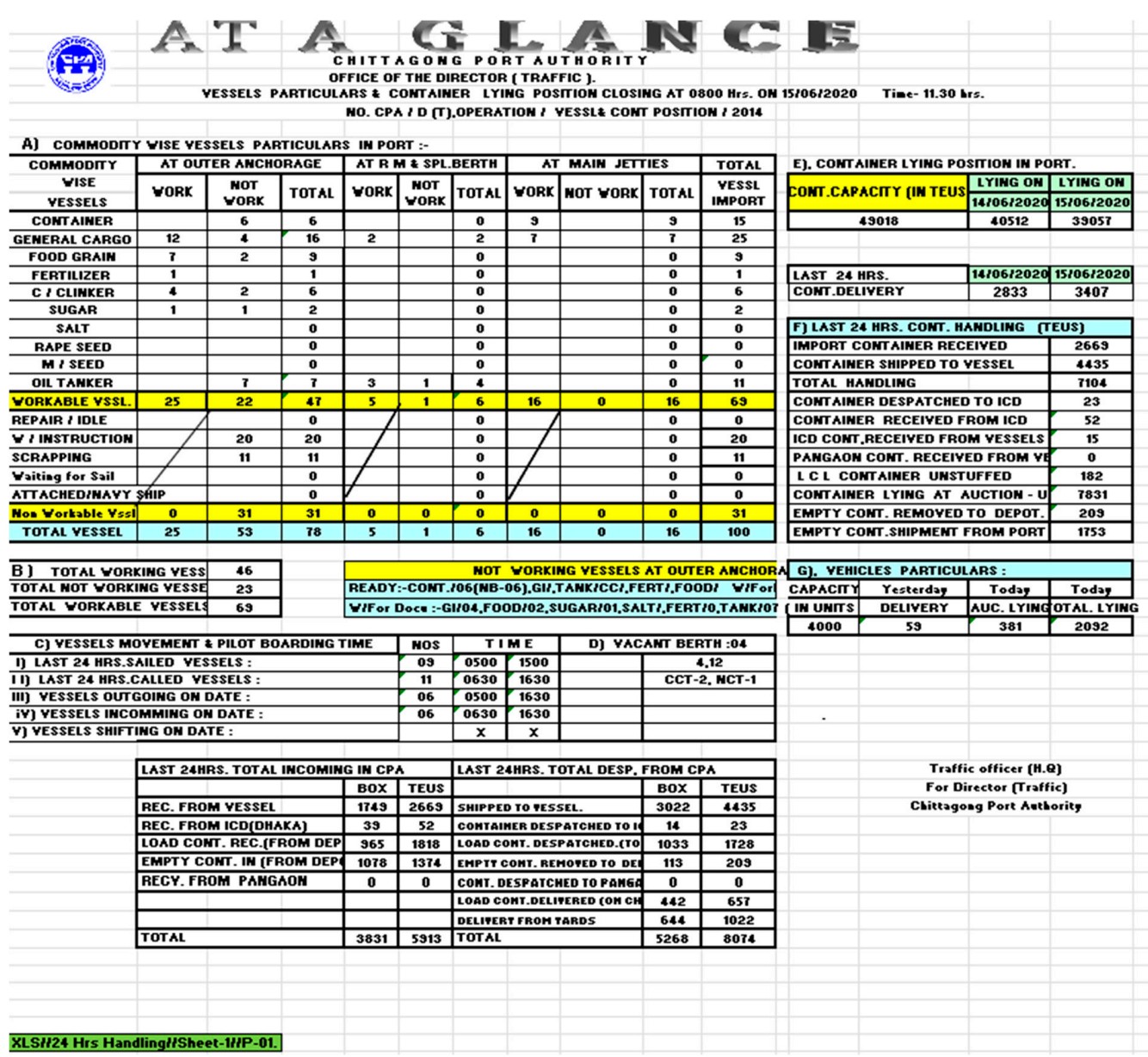

**Figure A2.** Vessel particulars and Yard Utilisation Data Sheet.

# Appendix C

**Table A1.** Example simulation table from Excel.

| Random Number | Waiting Time in Days | Random Number | Berthing Capacity | Ships at Berth | Newly Berthed Ships | Total Ship at Berth | Random Number | Berthing Duration in Days | Crane Output | Container Received | Containers Departing | Containers Added to Yard | Other Equipment Availability | Random Number | Yard Duration Other than Customs | Customs Duration | Trucking and Gate Duration | Total Lead Time | Annual Throughput (Import) in Million TEUs | MFO-Marine | Diesel—Terminal | Diesel—Gate | CO$_2$ MFO-Marine | CO$_2$ Diesel—Terminal | CO$_2$ Diesel—Gate | CO$_2$—Total in Tons | CO$_2$—Total/TEU |
|---|---|---|---|---|---|---|---|---|---|---|---|---|---|---|---|---|---|---|---|---|---|---|---|---|---|---|---|
| 59 | 6 | 8 | 1 | 7 | 0 | 7 | 89 | 3.5 | 48 | 6600 | 4800 | 1800 | 45 | 85 | 9 | 1 | 1 | 21 | 2.31 | 21,240.98 | 14,376.67 | 17,640 | 68,098.58 | 44,768.95 | 54,930.96 | 167,798.49 | 72.64 |
| 76 | 7 | 28 | 2 | 6 | 0 | 6 | 37 | 1.5 | 48 | 5808 | 4800 | 2808 | 45 | 25 | 3 | 1 | 1 | 14 | 2.03 | 19,140.98 | 14,376.67 | 17,640 | 61,365.98 | 44,768.95 | 54,930.96 | 161,065.89 | 79.23 |
| 69 | 7 | 1 | 1 | 7 | 1 | 8 | 45 | 1.5 | 48 | 6600 | 4800 | 4608 | 45 | 6 | 3 | 1 | 1 | 14 | 2.31 | 19,140.98 | 14,376.67 | 17,640 | 61,365.98 | 44,768.95 | 54,930.96 | 161,065.89 | 69.73 |
| 86 | 8 | 93 | 2 | 6 | 2 | 8 | 40 | 1.5 | 48 | 5808 | 4800 | 5616 | 45 | 17 | 3 | 1 | 1 | 15 | 2.03 | 21,240.98 | 14,376.67 | 17,640 | 68,098.58 | 44,768.95 | 54,930.96 | 167,798.49 | 82.55 |
| 75 | 7 | 42 | 2 | 6 | 2 | 8 | 23 | 0.5 | 48 | 5808 | 4800 | 6624 | 45 | 60 | 6 | 1 | 1 | 16 | 2.03 | 17,040.98 | 14,376.67 | 17,640 | 54,633.38 | 44,768.95 | 54,930.96 | 154,333.29 | 75.92 |
| 70 | 7 | 89 | 2 | 6 | 1 | 7 | 12 | 0.5 | 48 | 5808 | 4800 | 7632 | 45 | 25 | 3 | 1 | 1 | 13 | 2.03 | 17,040.98 | 14,376.67 | 17,640 | 54,633.38 | 44,768.95 | 54,930.96 | 154,333.29 | 75.92 |
| 59 | 6 | 23 | 1 | 7 | 0 | 7 | 18 | 0.5 | 48 | 6600 | 4800 | 9432 | 45 | 84 | 9 | 1 | 1 | 18 | 2.31 | 14,940.98 | 14,376.67 | 17,640 | 47,900.78 | 44,768.95 | 54,930.96 | 147,600.69 | 63.90 |
| 36 | 3 | 57 | 2 | 6 | 1 | 7 | 49 | 1.5 | 48 | 5808 | 4800 | 10,440 | 45 | 42 | 6 | 1 | 1 | 13 | 2.03 | 10,740.98 | 14,376.67 | 17,640 | 34,435.58 | 44,768.95 | 54,930.96 | 134,135.49 | 65.99 |
| 52 | 4 | 17 | 1 | 7 | 0 | 7 | 20 | 0.5 | 48 | 6600 | 4800 | 12,240 | 45 | 35 | 3 | 1 | 1 | 10 | 2.31 | 10,740.98 | 14,376.67 | 17,640 | 34,435.58 | 44,768.95 | 54,930.96 | 134,135.49 | 58.07 |
| 98 | 9 | 42 | 2 | 6 | 2 | 8 | 5 | 0.5 | 48 | 5808 | 4800 | 13,248 | 45 | 28 | 3 | 1 | 1 | 15 | 2.03 | 21,240.98 | 14,376.67 | 17,640 | 68,098.58 | 44,768.95 | 54,930.96 | 167,798.49 | 82.55 |
| 36 | 3 | 11 | 1 | 7 | 1 | 8 | 11 | 0.5 | 48 | 6600 | 4800 | 15,048 | 45 | 89 | 9 | 1 | 1 | 15 | 2.31 | 8640.98 | 14,376.67 | 17,640 | 27,702.98 | 44,768.95 | 54,930.96 | 127,402.89 | 55.15 |
| 46 | 4 | 75 | 2 | 6 | 2 | 8 | 35 | 1.5 | 48 | 5808 | 4800 | 16,056 | 45 | 10 | 3 | 1 | 1 | 11 | 2.03 | 12,840.98 | 14,376.67 | 17,640 | 41,168.18 | 44,768.95 | 54,930.96 | 140,868.09 | 69.30 |
| 89 | 8 | 55 | 2 | 6 | 2 | 8 | 32 | 1.5 | 48 | 5808 | 4800 | 17,064 | 45 | 62 | 6 | 1 | 1 | 18 | 2.08 | 21,240.98 | 14,376.67 | 17,640 | 68,098.58 | 44,768.95 | 54,930.96 | 167,798.49 | 82.55 |
| 41 | 3 | 47 | 2 | 6 | 2 | 8 | 29 | 1.5 | 48 | 5808 | 4800 | 18,072 | 45 | 39 | 3 | 1 | 1 | 10 | 2.03 | 10,740.98 | 14,376.67 | 17,640 | 34,435.58 | 44,768.95 | 54,930.96 | 134,135.49 | 65.99 |
| 95 | 8 | 75 | 2 | 6 | 2 | 8 | 77 | 3.5 | 48 | 5808 | 4800 | 19,080 | 45 | 74 | 6 | 1 | 1 | 20 | 2.03 | 25,440.98 | 14,376.67 | 17,640 | 81,563.78 | 44,768.95 | 54,930.96 | 181,263.69 | 89.17 |
| 38 | 3 | 94 | 4 | 4 | 3 | 7 | 70 | 2.5 | 48 | 4224 | 4800 | 18,504 | 45 | 67 | 6 | 1 | 1 | 14 | 1.47 | 12,840.98 | 14,376.67 | 17,640 | 41,168.18 | 44,768.95 | 54,930.96 | 140,868.09 | 95.28 |
| 46 | 4 | 84 | 2 | 6 | 2 | 8 | 34 | 1.5 | 48 | 5808 | 4800 | 19,512 | 45 | 31 | 3 | 1 | 1 | 11 | 2.03 | 12,840.98 | 14,376.67 | 17,640 | 41,168.18 | 44,768.95 | 54,930.96 | 140,868.09 | 69.30 |
| 99 | 10 | 42 | 2 | 6 | 2 | 8 | 27 | 1.5 | 48 | 5808 | 4800 | 20,520 | 45 | 28 | 3 | 1 | 1 | 17 | 2.03 | 25,440.98 | 14,376.67 | 17,640 | 81,563.78 | 44,768.95 | 54,930.96 | 181,263.69 | 89.17 |
| 64 | 6 | 26 | 2 | 6 | 2 | 8 | 93 | 3.5 | 48 | 5808 | 4800 | 21,528 | 45 | 69 | 6 | 1 | 1 | 18 | 2.03 | 21,240.98 | 14,376.67 | 17,640 | 68,098.58 | 44,768.95 | 54,930.96 | 167,798.49 | 82.55 |
| 36 | 3 | 63 | 2 | 6 | 2 | 8 | 35 | 1.5 | 48 | 5808 | 4800 | 22,536 | 45 | 52 | 6 | 1 | 1 | 13 | 2.03 | 10,740.98 | 14,376.67 | 17,640 | 34,435.58 | 44,768.95 | 54,930.96 | 134,135.49 | 65.99 |
| 91 | 8 | 98 | 4 | 4 | 4 | 8 | 3 | 0.5 | 48 | 4224 | 4800 | 21,960 | 45 | 8 | 3 | 1 | 1 | 14 | 1.47 | 19,140.98 | 14,376.67 | 17,640 | 61,365.98 | 44,768.95 | 54,930.96 | 161,065.89 | 108.95 |
| 97 | 9 | 88 | 2 | 6 | 2 | 8 | 44 | 1.5 | 48 | 5808 | 4800 | 22,968 | 45 | 20 | 3 | 1 | 1 | 16 | 2.03 | 23,340.98 | 14,376.67 | 17,640 | 74,831.18 | 44,768.95 | 54,930.96 | 174,531.09 | 85.86 |
| 71 | 7 | 14 | 1 | 7 | 1 | 8 | 60 | 2.5 | 48 | 6600 | 4800 | 24,768 | 45 | 72 | 6 | 1 | 1 | 18 | 2.31 | 21,240.98 | 14,376.67 | 17,640 | 68,098.58 | 44,768.95 | 54,930.96 | 167,798.49 | 72.64 |
| 9 | 1 | 42 | 2 | 6 | 1 | 7 | 0 | 0.5 | 48 | 5808 | 4800 | 25,776 | 45 | 18 | 3 | 1 | 1 | 7 | 2.03 | 4440.98 | 14,376.67 | 17,640 | 14,237.78 | 44,768.95 | 54,930.96 | 113,937.69 | 56.05 |
| 21 | 2 | 40 | 2 | 6 | 2 | 8 | 65 | 2.5 | 48 | 5808 | 4800 | 26,784 | 45 | 91 | 9 | 1 | 1 | 16 | 2.03 | 10,740.98 | 14,376.67 | 17,640 | 34,435.58 | 44,768.95 | 54,930.96 | 134,135.49 | 65.99 |
| 93 | 8 | 35 | 2 | 6 | 2 | 8 | 68 | 2.5 | 48 | 5808 | 4800 | 27,792 | 45 | 33 | 3 | 1 | 1 | 16 | 2.03 | 23,340.98 | 14,376.67 | 17,640 | 74,831.18 | 44,768.95 | 54,930.96 | 174,531.09 | 85.86 |
| 80 | 7 | 78 | 2 | 6 | 2 | 8 | 80 | 3.5 | 48 | 5808 | 4800 | 28,800 | 45 | 67 | 6 | 1 | 1 | 19 | 2.03 | 23,340.98 | 14,376.67 | 17,640 | 74,831.18 | 44,768.95 | 54,930.96 | 174,531.09 | 85.86 |
| 4 | 0 | 46 | 2 | 6 | 2 | 8 | 49 | 1.5 | 48 | 5808 | 4800 | 29,808 | 45 | 48 | 6 | 1 | 1 | 10 | 2.03 | 4440.98 | 14,376.67 | 17,640 | 14,237.78188 | 44,768.95038 | 54,930.96 | 113,937.6923 | 56.05 |
| 55 | 5 | 47 | 2 | 6 | 2 | 8 | 54 | 2.5 | 48 | 5808 | 4800 | 30,816 | 45 | 33 | 3 | 1 | 1 | 13 | 2.03 | 17,040.98 | 14,376.67 | 17,640 | 54,633.38188 | 44,768.95038 | 54,930.96 | 154,333.2923 | 75.92 |
| 93 | 8 | 70 | 2 | 6 | 2 | 8 | 28 | 1.5 | 48 | 5808 | 4800 | 31,824 | 45 | 37 | 3 | 1 | 1 | 15 | 2.03 | 21,240.98 | 14,376.67 | 17,640 | 68,098.58188 | 44,768.95038 | 54,930.96 | 167,798.4923 | 82.55 |

Waiting time:
=IF((IF(G15/100<$AJ$5,$AG$5,IF(G15/100<$AJ$6,$AG$6,IF(G15/100<$AJ$7,$AG$7,IF(G15/100<$AJ$8,$AG$8,IF(G15/100<$AJ$9,$AG$9,IF(G15/100<$AJ$10,$AG$10,IF(G15/100<$AJ$11,$AG$11,IF(G15/100<$AJ$12,$AG$12,$AG$13))))))))))

Berthing capacity:
=IF(I16/100<$AJ$45,$AG$45,IF(I16/100<$AJ$46,$AG$46,IF(I16/100<$AJ$47,$AG$47,IF(I16/100<$AJ$48,$AG$48,IF(I16/100<$AJ$49,$AG$49,$AG$50)))))

Berthing duration:
=IF((IF($Q$12<20,IF(N15/100<$AJ$29,$AG$29,IF(N15/100<$AJ$30,$AG$30,$AG$31)),IF(N15/100<$AJ$29,$AG$29,IF(N15/100<$AJ$30,$AG$30,$AG$31)))

Yard duration:
=IF(U15/100<$AJ$39,$AG$39,IF(U15/100<$AJ$40,$AG$40,$AG$41))

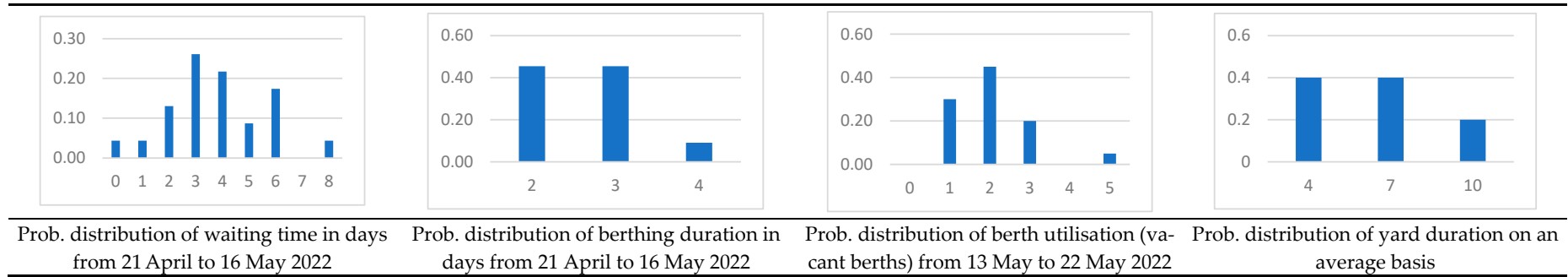

| Prob. distribution of waiting time in days from 21 April to 16 May 2022 | Prob. distribution of berthing duration in days from 21 April to 16 May 2022 | Prob. distribution of berth utilisation (vacant berths) from 13 May to 22 May 2022 | Prob. distribution of yard duration on an average basis |

**Figure A3.** Simulation MS Excel Formulae.

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
