# Peer review of "A Sustainable Port-Hinterland Container Transport System: The Simulation-Based Scenarios for CO2 Emission Reduction"

_sustainability, doi:10.3390/su15129444_

Round 1

Reviewer 1 Report

This paper calculates the CO2 emission reduction in the port of Chittagong in Bangladesh. The calculation is based on three scenarios implementation. An interesting research paper with data for port activities gathered and information regarding connection with the hinterland.

The paper could add other, previous studies that have dealt with the issue of CO2 emission in port areas (section 2.1). It seems that the next section (2.2) is a bit more in detail that section 2.1.

In section 2.2, third line, “… actions to be undertaken These methods…”, add a full stop after “undertaken”.

Page 4, “The gCO2/ton-km emission suggested by McKinnon and Piecyk [21] and Wang, Peng [8] lies within the range suggested by IPCC [18] for IWT.”. In Table 1, there is no emission factor for IWT from IPCC. So, what is written from the authors is not clear.

In Figure 1, please write the full name of FCL, LCL and CFS (maybe add a note?).

Table 4, correct TUEs (in third line of the table) with TEUs. In Straddle carrier (02 High) specify the 02nos. Also, the blank cell in Forklift truck (42-ton capacity) and Container Mover (50-ton capacity) for 2021, what does it mean? There is no such equipment anymore?

Page 11, three lines before the end of the page, “… the terminal handling CO2 emission is 26.56Kg per TEU…”, where is this number in Table 6?

In the scenarios analysis (e.g., page 14 for scenario III), can the authors describe how could yard operating time be achieved?

In the conclusions section, how this research could be of benefit for practitioners (or even other researchers)?

I have detected no serious editing in respect to the English language.

Author Response

Reviewer #1

Overall comment: “This paper calculates the CO2 emission reduction in the port of Chittagong in Bangladesh. The calculation is based on three scenarios implementation. An interesting research paper with data for port activities gathered and information regarding connection with the hinterland.”

Response: Thank you for summarizing this paper clearly and recognizing the significance of this paper to the existing literature on port-hinterland activities. Based on your comments, we have carried out a major revision to improve the quality of this research paper. We hope that the revised version has satisfactorily addressed your comments.  

  1. The paper could add other, previous studies that have dealt with the issue of CO2 emission in port areas (section 2.1). It seems that the next section (2.2) is a bit more in detail that section 2.1.”

Response: Thank you for bringing this issue to our attention. In the revised paper, we follow your suggestion and include more previous studies dealing with the issue of CO2 emissions in port areas to enrich the literature review section. Below are some major revisions and more detailed revisions can be found in the revised paper.

(Lines 33-39, Page 3) “Alternative energy and fuels are identified as an important influential factor for decreasing energy consumption and carbon emissions by IMO and other reviews [15]. Utilizing the renewable energy sources for port equipment operation is an increasing area of interest in the academic research nowadays [16]. On the other hand, the scheduling problems of port operating equipment are extensively observed in the existing literature. For example, carbon efficient scheduling of the rubber-tyred gantry cranes (RTGs) and electric rubber-tyred gantry cranes (ERTGs) is analysed by Chen and Zeng [17].”

(Lines 47-51, Page 3; Lines 1-3, Page 4) “The case of Qingdao port in China was analysed by Mamatok, Huang [21] through the implementation of scenario-based simulations. They argue that scenarios such as operating time optimization, spatial measures, equipment modernization and modal shift are important to CO2 emission reduction. The port-hinterland container transport system including three ports is analysed by Li, Kuang [22] through various scenarios on port selection based on new routes and multimodal transport as well as their impacts. In order to reduce carbon emissions in the port of Shenzhen, possible suggestions such as efficiency improvement of loading and unloading, connecting to shore power and low sulfur fuel oil use are proposed by Yang, Cai [23].”

  1. “In section 2.2, third line, “… actions to be undertaken These methods…”, add a full stop after “undertaken”.”

Response: Thank you for pointing out this error. A full stop has been added (Line 13, Page 4).

  1. “Page 4, “The gCO2/ton-km emission suggested by McKinnon and Piecyk [21] and Wang, Peng [8] lies within the range suggested by IPCC [18] for IWT.”. In Table 1, there is no emission factor for IWT from IPCC. So, what is written from the authors is not clear.”

Response: Thank you for raising your concern. As you may have noticed, the IPCC report does not provide a direct indication of IWT emissions. Instead, the range 10-40 provided in the report is a range for the entire shipping which includes coastal and international shipping. There is one category mentioned as ‘barge’ for which the emission factor is 50gCO2/ton-km, considering that the information is corrected in Table 1 (Pages 4-5) and both the IWT and coastal shipping along with a new category barge are merged together and added a new range 10-50.

  1. “In Figure 1, please write the full name of FCL, LCL and CFS (maybe add a note?).”

Response: Thank you for this reminder. The full names of FCL, LCL and CFS have been updated as “Full Container Load (FCL)”, “Less than Container Load (LCL)”, and “container freight station (CFS)” in the revised Figure 1 (Page 7).

  1. “Table 4, correct TUEs (in third line of the table) with TEUs. In Straddle carrier (02 High) specify the 02nos. Also, the blank cell in Forklift truck (42-ton capacity) and Container Mover (50-ton capacity) for 2021, what does it mean? There is no such equipment anymore.”

Response: Thank you for pointing out these errors. All of them have been corrected and a note has been added to Table 4 (Pages 7- 8) to explain the blank cells.

  1. “Page 11, three lines before the end of the page, “… the terminal handling CO2 emission is 26.56Kg per TEU…”, where is this number in Table 6.”

Response: Thank you for raising this point. The sub-total is added to Table 6 in the revised paper (Page 11).

  1. “In the scenarios analysis (e.g., page 14 for scenario III), can the authors describe how could yard operating time be achieved?”

Response: We apologize that the original paper did not clearly describe how to achieve a reduction in yard operating time under scenario III. To be honest, this paper is part of a whole doctoral thesis and the information on a reduction in yard operating time is provided in other sections. To avoid confusion, the following assumption has been added to the revised paper.

(Lines 32-39, Page 14) “It is observed from Figure 1 that it requires a container to stay in the yard for 10-12 days on average for customs and other inspections and finally unstuff and load in a trailer or truck or covered van for final delivery. Moreover, the improvement in the quay operation in scenario II requires a simultaneous operational improvement in the yard as well, otherwise, there will be congestion in the yard leading towards a deterioration of achievement in the quay operation. Therefore, in the third scenario an improvement in the yard operation (duration) and customs services is considered to understand the overall impact on the emission.”

  1. “In the conclusions section, how this research could be of benefit for practitioners (or even other researchers)?”

Response: Thank you for the comment. In the revised paper, the following content has been added to demonstrate how this research could be of benefit for both researchers and practitioners.

(Lines 25-35, Page 17) “For the case port-hinterland, this research first demonstrates the possibility of sustainable port-hinterland transport. It further indicates that the ports in developing countries like Bangladesh has much scope to concentrate in the operational efficiency as well as infrastructure development to achieve sustainability in the port-hinterland transport system. At the same time this research could benefit the researchers as well as practitioners explaining the implication of operational performance improvement and infrastructure development on the sustainable port-hinterland container transport system. The port operators, port authorities and policy makers can focus in these areas for CO2 emission reduction. Policy could cover focusing on the maximization of resource utilization and addition of equipment in the short term and gradually go for investment in additional infrastructure. Reduction in CO2 emissions from such initiatives can contribute to the air quality and liveability in the port-hinterland neighbourhood as well as port-city interaction and relationship.”

Reviewer 2 Report

This study estimated CO2 emissions from Dhaka-Chittagong port-hinterland transport system from port activity data and compared several management scenarios for their potential in CO2 emission reduction. The findings are valuable in supporting relevant decision making processes.  However, the authors should add substantially more information to the Methodology section to better clarify their calculation of CO2 emissions; and include key raw data (e.g., the type of fuels, the breakup of energy sources, and fuel efficiency) in the Appendix or supplemental material. An uncertainty analysis and a sensitivity analysis are strongly recommended, as well as a comparison of the port's CO2 emission data (kg CO2/TEU) with other ports' in the literature.

Overall the manuscript is easily readable. There are some minor issues with structuring, though. For example, the 2nd paragraph of Methodology apparently should belong to results and discussions.

Author Response

“This study estimated CO2 emissions from Dhaka-Chittagong port-hinterland transport system from port activity data and compared several management scenarios for their potential in CO2 emission reduction. The findings are valuable in supporting relevant decision making processes. However, the authors should add substantially more information to the Methodology section to better clarify their calculation of CO2 emissions; and include key raw data (e.g., the type of fuels, the breakup of energy sources, and fuel efficiency) in the Appendix or supplemental material. An uncertainty analysis and a sensitivity analysis are strongly recommended, as well as a comparison of the port's CO2 emission data (kg CO2/TEU) with other ports' in the literature.”

Response: Thank you for summarizing this paper clearly, recognizing its value added to the existing literature, and providing your constructive suggestions for improving the quality of this paper.

Regarding the calculation of CO2 emissions, we apologize that we didn’t clearly clarify this in the Methodology section in the original paper. However, you may have noticed that the required information on data sources were discussed throughout sections 2, 3 and 4 with consideration of data relevance. To better address your concern, we have made revisions in the following aspects: First, a reminder of data sources is given in Introduction. (Lines 12-13, Page 3) “It is worth mentioning that sections 2, 3 and 4 include a respective data description and discussion with consideration of data relevance to each section.” Second, the following sentences are added to supplement the previous discussion about the calculation of CO2 emissions in this paragraph. (Lines 15-17, Page 10) “All these data discussed in section 2.3 are collected from the CPA web sources and some missing data from the Port and Traffic Department of CPA specific discussion of which are included in the specific sections where it is used throughout this paper.” Third, section 3 has been renamed as “Methodology and Materials” in the revised paper to include data sources as well. Furthermore, to help the audience find the required information easily, an indication of the distribution of data is included in the revised section 3, such as “section 2.3” (Line 36, Page 9) and “section 2 (sections 2.3 and 2.4)” (Line 1, Page 10).

Considering that almost all the data are collected from CPA web sources, very few missing data are collected from hard materials published by the Port and Traffic Department of CPA rather than from their website. To include this information, these data are added as Appendices (Pages 22 and 23) in the revised paper. As for the detailed data on fuel type, fuel efficiency and the breakup of energy sources, they are not readily available from the published sources and they are normally provided as a whole, making it difficult to provide such details in this research. To reflect your concern, the limitation in Conclusion has been revised as (Lines 43-45, Page 17) “One potential limitation in for the emission calculation was poor data availability (e.g., fuel type, fuel efficiency and the breakup of energy sources) for the case rail and IWT transportation to calculate the fuel consumptions.” In addition, the data on other ports’ CO2 emissions are compared in Section 4. (Lines 13-22, Page 12) “There are not many similar studies in the literature, except for Huzaifi, Budiyanto [12] which found that carbon emissions in Jakarta International Container Terminal are 29.08 kg per TEU, but as mentioned earlier it considered the inward services from berthing to taking the containers out of port. Another study by Yang et al calculated that the per TEU CO2 emissions in the port of Shenzhen are 23.49 kg [22, 23]. Comparing the fuel consumption with previous empirical findings, noticeable differences can be seen, which are mainly due to the performance gap between Chittagong Port and the above ports. That is, the performance of Chittagong port in every aspect of waiting, berthing, dwell and gate operation time is not up to the mark. Therefore, policymakers have the opportunity from this calculation to look at the emissions status of the port and the performance issue from an environmental sustainability perspective.”

We carefully considered your suggestion about “an uncertainty analysis and a sensitivity analysis” but we decided not to include it in the revised paper due to the following reasons: First, as you may have noticed, some input variables of the simulation analysis are kept fixed whereas some are random. As shown in Appendix III, the random variables include an upper limit and a lower limit with the corresponding probability distribution, which, to a certain extent, demonstrate uncertainty. The simulation analysis has considered the above uncertainty by examining the impacts of both the lower and upper values for random input variables. However, the output variable is presented as an average instead of showing its lower and upper values, enabling us to better compare CO2 emissions before and after implementing an improvement. Second, the current simulations have analyzed how a change in a fixed input variable (e.g., working hours for the quay crane in Scenario I) can affect the output variable (e.g., per TEU emission), which is similar to a sensitivity analysis. In this sense, the current simulation analysis of three scenarios can well achieve the objective of this paper that is to help the port users, service providers and policymakers better understand how operational performance improvement and infrastructure development can reduce CO2 emissions for the case port-hinterland.

Reviewer 3 Report

This paper presents a study on CO2 emissions in the port-hinterland container transport system, focusing on the specific case of the Dhaka-Chittagong system in Bangladesh. This system represents a significant share of international container trade, with over 2 million twenty-foot equivalent units (TEUs) per year. The study proposes various emission reduction scenarios, including infrastructure development, operational performance improvement at the port, and modal shift for the hinterland. By analyzing current performance statistics and the feasibility of these scenarios, the researchers conclude that implementing these measures could significantly reduce CO2 emissions in the port-hinterland system, thus contributing to the fight against climate change.

"The data collection is a weak point in the article. The authors must clearly justify the methodology used to collect the data and also justify the study period."

What is the problem addressed in this article?

Explicitly demonstrate the originality of the article and its contribution to advancing knowledge.

In the conclusion, highlight the empirical implications of the research on the environment and society.

Provide recommendations regarding public policies that can be implemented to reduce CO2 emissions.

Author Response

Reviewer #3

Overall comment: “This paper presents a study on CO2 emissions in the port-hinterland container transport system, focusing on the specific case of the Dhaka-Chittagong system in Bangladesh. This system represents a significant share of international container trade, with over 2 million twenty-foot equivalent units (TEUs) per year. The study proposes various emission reduction scenarios, including infrastructure development, operational performance improvement at the port, and modal shift for the hinterland. By analyzing current performance statistics and the feasibility of these scenarios, the researchers conclude that implementing these measures could significantly reduce CO2 emissions in the port-hinterland system, thus contributing to the fight against climate change.”

Response: Thank you for summarizing this paper clearly and logically. Based on your constructive comments, we have carried out a thorough major revision to improve the quality of this paper. We hope that the revised version has satisfactorily addressed your concerns.

  1. “The data collection is a weak point in the article. The authors must clearly justify the methodology used to collect the data and also justify the study period.”

Response:  Thank you for pointing out this weakness and providing your suggestion. We apologize that we didn’t clearly clarify data collection and study period in the original paper. However, you may have noticed that the required information on data collection is discussed throughout sections 2, 3 and 4 with consideration of data relevance. To overcome this weakness, we have made revisions in the following aspects: First, a reminder of data sources is given in Introduction. (Lines 12-13, Page 3) “It is worth mentioning that sections 2, 3 and 4 include a respective data description and discussion with consideration of data relevance to each section.” Second, the following sentences are added to supplement the previous discussion about the calculation of CO2 emissions. (Lines 15-17, Page 10) “All these data discussed in section 2.3 are collected from the CPA web sources and some missing data from the Port and Traffic Department of CPA specific discussion of which are included in the specific sections where it is used throughout this paper.” Third, section 3 has been renamed as “Methodology and Materials” in the revised paper to include data sources as well. Furthermore, to help the audience find the required information easily, an indication of the distribution of data is included in the revised section 3, such as “section 2.3” (Line 36, Page 9) and “section 2 (sections 2.3 and 2.4)” (Line 1, Page 10). Fourth, considering that almost all the data are collected from CPA web sources, very few missing data are collected from hard materials published by the Port and Traffic Department of CPA rather than from their website. To include this information, these data are added as Appendices (Pages 22 and 23) in the revised paper. Finally, this research addresses the study period in different sections when introducing the used data. Considering that the objective of this research is to understand the impact of operational performance improvement and infrastructure development on the CO2 emission reduction, it first compares CO2 emissions based on the port performance data for the year 2020 and 2022, and then the other scenarios are designed and implemented. As a result, the study period is addressed in section 2.3 (Lines 24, 26, Pages 6; Line 7, Page 7; Lines 6, 10, Page 8), section 3 (Line 4, Page 10), and section 4 (Title of Table 6; Line 8, Page 11; Lines 23, 26, 30, Page 12; Lines 10, 13, 15, 22, Page 14).

  1. “What is the problem addressed in this article?”

Response:  Thank you for raising this issue. As you may have noticed, the research problem has been addressed in the Introduction section. (Lines 44-51, Page 2; Lines 1-5, Page 3) “As the 58th largest container port in the world and the main gateway port in Bangladesh in terms of throughput, Chittagong port handles around three million TEUs annually and more than 90% of Bangladesh’s containerized international trade. Notably, the Dhaka-Chittagong hinterland transports around 70% of these containers. As a result, any CO2 emission reduction in this case port-hinterland container transport system can significantly impact the local emission scenarios. On the other hand, the existing port related literature has not paid sufficient attention to port-hinterland from an environmental sustainability perspective. Referring to a recent study on the competitiveness of this port, users suggest that the environmental practices have the least competitive aspect for port attractiveness [8]. Therefore, the calculation of CO2 emissions and the proposed emission reduction solutions through the implementation of different scenarios can contribute to the existing port management literature from the environmental sustainability perspective. In addition, this research provides implications for examining port users’ CO2 emissions while doing their business, thereby contributing to port service providers and policymakers when formulating sustainability policies for port-hinterland transportation.”

  1. “Explicitly demonstrate the originality of the article and its contribution to advancing knowledge.”

Response:  Thank you for bringing this point to our attention. According to your comment, the following content has been added to demonstrate the originality and contributions of this paper. (Lines 25-30, Page 17) “For the case port-hinterland, this research first demonstrates the possibility of sustainable port-hinterland transport. It further indicates that the ports in developing countries like Bangladesh has much scope to concentrate in the operational efficiency as well as infrastructure development to achieve sustainability in the port-hinterland transport system. At the same time this research could benefit the researchers as well as practitioners explaining the implication of operational performance improvement and infrastructure development on the sustainable port-hinterland container transport system.”

  1. “In the conclusion, highlight the empirical implications of the research on the environment and society.”

Response:  Thank you for raising this issue. As suggested, the following content has been added to highlight the empirical implications of this research on the environment and society. (Lines 33-35, Page 17) “Reduction in CO2 emissions from such initiatives can contribute to the air quality and liveability in the port-hinterland neighborhood as well as port-city interaction and relationship.”

  1. “Provide recommendations regarding public policies that can be implemented to reduce CO2 emissions.”

Response:  According to your constructive comment, the following content has been added. (Lines 31-35, Page 17) “Policy could cover focusing on the maximization of resource utilization and addition of equipment in the short term and gradually go for investment in additional infrastructure. Reduction in CO2 emissions from such initiatives can contribute to the air quality and liveability in the port-hinterland neighborhood as well as port-city interaction and relationship.”

Round 2

Reviewer 2 Report

The authors have properly addressed my comments. I would recommend the acceptance of the manuscript for publication.

No further issue identified.

Author Response

Note: the attachment is the language editing invoice

Quality of English Language: “Minor editing of English language required.”

Response: Thank you for your comment. This study formed part of Mr. Khandaker Rasel Hasan’s PhD thesis and his thesis has been proofread by a professional Editor Bron Fionnachd-Féin ([email protected]). For your easy reference, we also submitted the language editing invoice in the submission system (Invoice no: KRH/5/23-1). We hope that the revised version has satisfactorily addressed your concerns.

The authors have properly addressed my comments. I would recommend the acceptance of the manuscript for publication.”

Response: Thank you for recognizing our efforts. Your constructive comments have significantly improved the quality of this manuscript.
